# Some Hesitant Fuzzy Hamacher Power-Aggregation Operators for Multiple-Attribute Decision-Making

**Mi Jung Son [1], Jin Han Park [2],\* and Ka Hyun Ko [2]**

[1] Department of Data Information, Korea Maritime and Ocean University, Busan 606-791, Korea
[2] Department of Applied Mathematics, Pukyong National University, Busan 608-737, Korea
\* Correspondence: jihpark@pknu.ac.kr; Tel.: +82-51-629-5530

**Abstract:** As an extension of the fuzzy set, the hesitant fuzzy set is used to effectively solve the hesitation of decision-makers in group decision-making and to rigorously express the decision information. In this paper, we first introduce some new hesitant fuzzy Hamacher power-aggregation operators for hesitant fuzzy information based on Hamacher *t*-norm and *t*-conorm. Some desirable properties of these operators is shown, and the interrelationships between them are given. Furthermore, the relationships between the proposed aggregation operators and the existing hesitant fuzzy power-aggregation operators are discussed. Based on the proposed aggregation operators, we develop a new approach for multiple-attribute decision-making problems. Finally, a practical example is provided to illustrate the effectiveness of the developed approach, and the advantages of our approach are analyzed by comparison with other existing approaches.

**Keywords:** hesitant fuzzy element (HFE); Hamacher operations; hesitant fuzzy Hamacher power-aggregation operators; multiple-attribute decision-making (MADM)

## 1. Introduction

Since the fuzzy set (FS) was introduced by Zadeh [1], it has received much attention for its applicability. Some classical extensions of the FS, such as the interval-valued fuzzy set (IVFS) [2], intuitionistic fuzzy set (IFS) [3], interval-valued intuitionistic fuzzy set (IVIFS) [4], type-2 fuzzy set (T2FS) [5], type-*n* fuzzy set (T*n*FS) [5], and fuzzy multiset (FMS) [6], were then developed. However, it is often faced with the fact that the difficulty of setting membership degree for an element in a set arises not from the possibility distribution of possible values (as in T2FS) or the margin of error (as in IVFS or IFS), but from the hesitation between several different values. The concept of hesitant fuzzy set (HFS) was introduced by Narukawa and Torra [7,8] to deal with such cases. The HFS has the advantage of representing the membership degree of one element to a set by a set of possible values between 0 and 1, so it is an effective tool to represent a decision-maker's hesitation in expressing his/her preferences for objects than the FS or its classical extensions. In this regard, the HFS theory has been applied to many practical applications such as decision-making [9–18].

The goal of multiple-attribute decision-making (MADM), based on preferences provided by the decision-makers, is to select the most desirable alternative(s) from a given set of feasible alternatives. MADM methods classified as conventional and fuzzy. The conventional MADM methods are seen inadequate to handle uncertainty in linguistic terms [19]. Hence, it is proposed to apply MADM methods with the FS and its extensions to cope with vagueness in a decision-making process. Furthermore, these fuzzy methods enable more concrete results. Besides, the FS and its extensions helps to decision-makers to express their opinions by means of linguistic terms. Therefore, more sensitive results can be obtained by applying fuzzy MADM methods to various science and engineering fields such as supplier selection and forecasting [20–23].

The aggregation operators are most commonly used as tools to combine each individual preference information into the overall preference information and to elicit collective preference value for each alternative. The power average (PA) and power-ordered weighted average (POWA), introduced by Yager [24], are the nonlinear weighted average aggregation tools whose weight vectors depend on the input arguments and allow the argument values to support each other [25]. In particular, compared to most aggregation operators, the PA and POWA operators have the advantage of incorporating information about the relationship between argument values that are combined. So these operators have received a lot of attention from researchers in recent years, particularly Xu and Yager [25], Zhou et al. [26] and Zhang [15] introduced some new power-aggregation operators, including the weighted generalizations of these operators. However, these power-aggregation operators only deal with arguments, which are exact numerical values.

In real life, we often face situations where input arguments are expressed not as exact numerical values but as interval numbers [27], intuitionistic fuzzy numbers [28–30], interval-valued intuitionistic fuzzy numbers [31], linguistic variables [32–34], uncertain linguistic variables [35–37], or hesitant fuzzy elements (HFEs) [10,11]. Many extensions of power-aggregation operators have been proposed to address these situations: the uncertain power-aggregation operators [25,38,39], intuitionistic fuzzy power-aggregation operators [26,40], interval-valued intuitionistic fuzzy power-aggregation operators [40], linguistic power-aggregation operators [41–44] and hesitant fuzzy power-aggregation operators [15,18]. In particular, with respect to HFEs, Zhang [15] proposed a family of hesitant fuzzy power-aggregation operators, including the hesitant fuzzy power-weighted average/geometric (HFPWA or HFPWG), generalized hesitant fuzzy power-weighted average/geometric (GHFPWA or GHFPWG), hesitant fuzzy power-ordered weighted average/geometric (HFPOWA or HFPOWG), and generalized hesitant fuzzy power-ordered weighted average/geometric (GHFPOWA or GHFPOWG) operators, and applied them to solve multiple criteria group decision-making problems under hesitant fuzzy environment.

It is worthwhile to mention that operational rules play a key role in integrating information using power-aggregation operators. A lot of research about power-aggregation operators for the FS and its extensions has been done by operational rules using various pairs of triangular norm (shortly *t*-norm) and triangular conorm (shortly *t*-conorm) [45] in recent years. The aforementioned hesitant fuzzy power-aggregation operators, such as the HFPWA, HFPWG, GHFPWA, GHFPWG, HFPOWA, HFPOWG, GHFPOWA, and GHFPOWG operators, are based on the algebraic product and algebraic sum operational rules on HFEs, which are a pair of the special dual *t*-norm and *t*-conorm [45]. The algebraic product and algebraic sum are the basic operations on HFEs, they are not the only ones. The Einstein *t*-norm and *t*-conorm, as another pair of special *t*-norm and dual *t*-conorm, are alternatives to the algebraic product and algebraic sum, respectively, for operational rules on HFEs. Yu [16] extended the Einstein *t*-norm and *t*-conorm to HFEs, and developed some hesitant fuzzy Einstein aggregation operators based on the Einstein product and Einstein sum operational rules on HFEs. By mean of these operational rules on HFEs, Yu et al. [18] proposed a wide range of hesitant fuzzy power-aggregation operators, such as the hesitant fuzzy Einstein power-weighted average/geometric (HFEPWA or HFEPWG), generalized hesitant fuzzy Einstein power-weighted average/geometric (GHFEPWA or GHFEPWG), hesitant fuzzy Einstein power-ordered weighted average/geometric (HFEPOWA or HFEPOWG), and generalized hesitant fuzzy Einstein power-ordered weighted average/geometric (GHFEPOWA or GHFEPOWG) operators, and applied them to deal with MADM with hesitant fuzzy information. Hamacher [46] proposed a more generalized *t*-norm and *t*-conorm, called the Hamacher *t*-norm and *t*-conorm. These Hamacher *t*-norm and *t*-conorm are more general and flexible because they are a generalization of the algebraic *t*-norm and *t*-conorm and the Einstein *t*-norm and *t*-conorm [46]. Tan et al. [17] gave some operations on HFEs based on Hamacher *t*-norm and *t*-conorm, and developed some hesitant fuzzy Hamacher aggregation operators. In this paper, by means of Hamacher operations on HFEs, we propose a family of hesitant fuzzy Hamacher power-aggregation operators that allow decision-makers to have more

choices in MADM problems. This study is very necessary because it is an integrated treatment of works by Zhang [15] and Yu et al. [18].

To do so, this paper is organized as follows: In Section 2, some basic concepts and notions of HFSs and Hamacher operations of HFEs based on the Hamacher *t*-norm and *t*-conorm are reviewed. Some results of Hamacher operations of HFEs are investigated. In Section 3, we present a wide range of hesitant fuzzy Hamacher power-aggregation operators for hesitant fuzzy information, some of their basic properties are discussed, and the relationships between the proposed operators and the existing hesitant fuzzy aggregation operators are investigated. In Section 4, we apply the proposed operators to develop an approach to MADM with hesitant fuzzy information. An example application of the new approach is provided, and a comparison with other hesitant fuzzy MADM approaches is performed. Some concluding remarks is given in Section 5.

## 2. Basic Concepts and Operations

### 2.1. Triangular Norms and Conorms

The operators play an important role in the beginning of FS theory. The *t*-norm and *t*-conorm used to define generalized union and intersection, respectively, is one of the important concepts in FS theory. They are defined as follows.

**Definition 1.** *[45] A triangular norm (t-norm) is a binary operation $T$ on the unit interval $[0,1]$, i.e., a function $T : [0,1] \times [0,1] \to [0,1]$, such that for all $x, y, z \in [0,1]$, the following four axioms are satisfied:*
*(1) (Boundary condition) $T(1,x) = x$;*
*(2) (Commutativity) $T(x,y) = T(y,x)$;*
*(3) (Associativity) $T(x,T(y,z)) = T(T(x,y),z)$;*
*(4) (Monotonicity) $T(x_1,y_1) \leq T(x_2,y_2)$ if $x_1 \leq x_2$ and $y_1 \leq y_2$.*
*The corresponding triangular conorm (t-conorm) of $T$ (or the dual of $T$) is the function $S : [0,1] \times [0,1] \to [0,1]$ defined by $S(x,y) = 1 - T(1-x,1-y)$ for each $x, y \in [0,1]$.*

Among many *t*-norms and *t*-conorms, there are the following basic *t*-norms and *t*-conorms: minimum $T_M$ and maximum $S_M$, algebraic product $T_A$ and algebraic sum $S_A$, Einstein product $T_E$ and Einstein sum $S_E$, bounded difference $T_B$ and bounded sum $S_B$, and drastic product $T_D$ and drastic sum $S_D$, given respectively as follows:

- $T_M(x,y) = \min(x,y)$, $S_M(x,y) = \max(x,y)$;
- $T_A(x,y) = xy$, $S_A(x,y) = x + y - xy$;
- $T_E(x,y) = \dfrac{xy}{1 + (1-x)(1-y)}$, $S_E(x,y) = \dfrac{x+y}{1+xy}$;
- $T_B(x,y) = \max(0, x+y-1)$, $S_B(x,y) = \min(1, x+y)$;
- $T_D(x,y) = \begin{cases} 0, & \text{if } (x,y) \in [0,1)^2 \\ \min(x,y), & \text{otherwise} \end{cases}$, $S_D(x,y) = \begin{cases} 1, & \text{if } (x,y) \in (0,1]^2 \\ \max(x,y), & \text{otherwise.} \end{cases}$

These *t*-norms and *t*-conorms are ordered as follows:

$$T_D \leq T_B \leq T_E \leq T_A \leq T_M, \text{ and } S_M \leq S_A \leq S_E \leq S_B \leq S_D. \tag{1}$$

From (1), since the drastic product $T_D$ and minimum $T_M$ are the smallest and the largest *t*-norms, respectively, we know that $T_D \leq T \leq T_M$ for any *t*-norm $T$. In particular, the algebraic product $T_A$ and the Einstein product $T_E$ are two prototypic examples of the class of strict Archimedean *t*-norms [45].

Hamacher [46] proposed, as more generalized *t*-norm and *t*-conorm, the Hamacher *t*-norm and *t*-conorm as follows:

$$T_H^\zeta(x,y) = \frac{xy}{\zeta + (1-\zeta)(x+y-xy)}, \quad S_H^\zeta(x,y) = \frac{x+y-xy-(1-\zeta)xy}{\zeta + (1-\zeta)xy}, \quad \zeta > 0. \tag{2}$$

From (2), when $\zeta = 1$, then the Hamacher *t*-norm and *t*-conorm reduce to the algebraic *t*-norm $T_A$ and *t*-conorm $S_A$, respectively; when $\zeta = 2$, then then the Hamacher *t*-norm and *t*-conorm reduce to the Einstein *t*-norm $T_E$ and *t*-conorm $S_E$, respectively.

## 2.2. Hesitant Fuzzy Sets and Hesitant Fuzzy Elements

In the following, some basic concepts of hesitant fuzzy set and hesitant fuzzy element are briefly reviewed [7,8,10].

**Definition 2.** *[7,8] Let X be a fixed set, a hesitant fuzzy set (HFS) on X is defined in terms of function h that returns a subset of* $[0,1]$ *when applied to X. The HFS can be represented as the following mathematical symbol:*

$$E = \{\langle x, h_E(x)\rangle | x \in X\}, \tag{3}$$

*where* $h_E(x)$ *is a set of values in* $[0,1]$ *that denote the possible membership degrees of the element* $x \in X$ *to the set E. For convenience, we refer to* $h = h_E(x)$ *as a hesitant fuzzy element (HFE) and to H the set of all HFEs.*

Given three HFEs $h$, $h_1$ and $h_2$, Torra and Narukawa [7,8] and Xia and Xu [10] defined the following HFE operations:

(1) $h^c = \cup_{\gamma \in h}\{1 - \gamma\}$;

(2) $h_1 \cup h_2 = \cup_{\gamma_1 \in h_1, \gamma_2 \in h_2}\{\gamma_1 \vee \gamma_2\}$;

(3) $h_1 \cap h_2 = \cup_{\gamma_1 \in h_1, \gamma_2 \in h_2}\{\gamma_1 \wedge \gamma_2\}$;

(4) $h^\lambda = \cup_{\gamma \in h}\{\gamma^\lambda\}, \lambda > 0$;

(5) $\lambda h = \cup_{\gamma \in h}\{1 - (1-\gamma)^\lambda\}, \lambda > 0$;

(6) $h_1 \oplus h_2 = \cup_{\gamma_1 \in h_1, \gamma_2 \in h_2}\{\gamma_1 + \gamma_2 - \gamma_1\gamma_2\}$;

(7) $h_1 \otimes h_2 = \cup_{\gamma_1 \in h_1, \gamma_2 \in h_2}\{\gamma_1\gamma_2\}$.

Xia and Xu [10] also defined the following comparison rules for HFEs:

**Definition 3.** *[10] For a HFE h,* $s(h) = \frac{\sum_{\gamma \in h} \gamma}{l(h)}$ *is called the score function of h, where* $l(h)$ *is the number of elements in h. For two HFEs* $h_1$ *and* $h_2$*,*

- *if* $s(h_1) > s(h_2)$*, then* $h_1$ *is superior to* $h_2$*, denoted by* $h_1 > h_2$*;*
- *if* $s(h_1) = s(h_2)$*, then* $h_1$ *is indifferent to* $h_2$*, denoted by* $h_1 = h_2$*.*

Let $h_1$ and $h_2$ be two HFEs. In the most case, $l(h_1) \neq l(h_2)$; for convenience, let $l = \max\{l(h_1), l(h_2)\}$. To compare $h_1$ and $h_2$, Xu and Xia [11] extended the shorter HFE until the length of both HFEs was the same. The simplest way to extend the shorter HFE is to add the same value repeatedly. In fact, we can extend the shorter ones by adding any values in them. The selection of these values mainly depends on the decision-makers' risk preferences. Optimists anticipate desirable outcomes and may add the maximum value, while pessimists expect unfavorable outcomes and may add the minimum value [11]. In this paper, we assume that the decision-makers are all pessimistic (other situation can also be studied similarly).

Xu and Xia [11] proposed various distance measures for HFEs, including the hesitant normalized Hamming distance defined as follows:

$$d(h_1, h_2) = \frac{1}{l} \sum_{i=1}^{l} \left| h_1^{\sigma(i)} - h_2^{\sigma(i)} \right|, \tag{4}$$

where $h_1^{\sigma(i)}$ and $h_2^{\sigma(i)}$ are the $i$th largest values in $h_1$ and $h_2$, respectively.

Intrinsically, the addition and multiplication operators proposed by Xia and Xu [10] are algebraic sum and algebraic product operational rules on HFEs, respectively, and are a special pair of dual $t$-norm and $t$-conorm. Recently, Tan et al. [17] extended these operations to obtain more general operations on HFEs by means of the Hamacher $t$-norm and $t$-conorm as follows:

**Definition 4.** [17] *For any given three HFEs $h$, $h_1$, $h_2$, and $\zeta > 0$, the Hamacher operations on HFEs are defined as follows:*

*(1)* $h_1 \oplus_H h_2 = \cup_{\gamma_1 \in h_1, \gamma_2 \in h_2} \left\{ \frac{\gamma_1 + \gamma_2 - \gamma_1\gamma_2 - (1-\zeta)\gamma_1\gamma_2}{1 - (1-\zeta)\gamma_1\gamma_2} \right\}$;

*(2)* $h_1 \otimes_H h_2 = \cup_{\gamma_1 \in h_1, \gamma_2 \in h_2} \left\{ \frac{\gamma_1\gamma_2}{\zeta + (1-\zeta)(\gamma_1 + \gamma_2 - \gamma_1\gamma_2)} \right\}$;

*(3)* $\lambda \cdot_H h = \cup_{\gamma \in h} \left\{ \frac{(1+(\zeta-1)\gamma)^\lambda - (1-\gamma)^\lambda}{(1+(\zeta-1)\gamma)^\lambda + (\zeta-1)(1-\gamma)^\lambda} \right\}, \lambda > 0$;

*(4)* $h^{\wedge_H \lambda} = \cup_{\gamma \in h} \left\{ \frac{\zeta\gamma^\lambda}{(1+(\zeta-1)(1-\gamma))^\lambda + (\zeta-1)\gamma^\lambda} \right\}, \lambda > 0$.

In particular, if $\zeta = 1$, then these operations on HFEs reduce to those proposed by Xia and Xu [10]; if $\zeta = 2$, then these operations on HFEs reduce to the following:

*(1)* $h_1 \oplus_\varepsilon h_2 = \cup_{\gamma_1 \in h_1, \gamma_2 \in h_2} \left\{ \frac{\gamma_1 + \gamma_2}{1 - \gamma_1\gamma_2} \right\}$;

*(2)* $h_1 \otimes_\varepsilon h_2 = \cup_{\gamma_1 \in h_1, \gamma_2 \in h_2} \left\{ \frac{\gamma_1\gamma_2}{1 + (1-\gamma_1)(1-\gamma_2)} \right\}$;

*(3)* $\lambda \cdot_\varepsilon h = \cup_{\gamma \in h} \left\{ \frac{(1+\gamma)^\lambda - (1-\gamma)^\lambda}{(1+\gamma)^\lambda + (1-\gamma)^\lambda} \right\}, \lambda > 0$;

*(4)* $h^{\wedge_\varepsilon \lambda} = \cup_{\gamma \in h} \left\{ \frac{2\gamma^\lambda}{(2-\gamma)^\lambda + \gamma^\lambda} \right\}, \lambda > 0$,

which are defined as Einstein operations on HFEs by Yu [16].

**Theorem 1.** *Let $h$, $h_1$ and $h_2$ be three HFEs, $\lambda > 0$, $\lambda_1 > 0$ and $\lambda_2 > 0$, then*

*(1)* $h_1 \oplus_H h_2 = h_2 \oplus_H h_1$;

*(2)* $h \oplus_H (h_1 \oplus_H h_2) = (h \oplus_H h_1) \oplus_H h_2$;

*(3)* $\lambda_1 \cdot_H (\lambda_2 \cdot_H h) = (\lambda_1 \lambda_2) \cdot_H h$;

*(4)* $\lambda \cdot_H (h_1 \oplus_H h_2) = (\lambda \cdot_H h_1) \oplus_H (\lambda \cdot_H h_2)$;

*(5)* $h_1 \otimes_H h_2 = h_2 \otimes_H h_1$;

*(6)* $h \otimes_H (h_1 \otimes_H h_2) = (h \otimes_H h_1) \otimes_H h_2$;

*(7)* $(h_1 \otimes_H h_2)^{\wedge_H \lambda} = h_1^{\wedge_H \lambda} \otimes_H h_2^{\wedge_H \lambda}$;

*(8)* $(h^{\wedge_H \lambda_1})^{\wedge_H \lambda_2} = h^{\wedge_H (\lambda_1 \lambda_2)}$.

**Proof.** Since (1), (2), (5) and (6) are trivial, we prove (3), (4), (7) and (8).

(3) Since $\lambda_2 \cdot_H h = \cup_{\gamma \in h} \left\{ \frac{(1+(\zeta-1)\gamma)^{\lambda_2} - (1-\gamma)^{\lambda_2}}{(1+(\zeta-1)\gamma)^{\lambda_2} + (\zeta-1)(1-\gamma)^{\lambda_2}} \right\}$, then we have

$\lambda_1 \cdot_H (\lambda_2 \cdot_H h)$

$= \cup_{\gamma \in h} \left\{ \frac{\left(1 + (\zeta-1)\frac{(1+(\zeta-1)\gamma)^{\lambda_2} - (1-\gamma)^{\lambda_2}}{(1+(\zeta-1)\gamma)^{\lambda_2} + (\zeta-1)(1-\gamma)^{\lambda_2}}\right)^{\lambda_1} - \left(1 - \frac{(1+(\zeta-1)\gamma)^{\lambda_2} - (1-\gamma)^{\lambda_2}}{(1+(\zeta-1)\gamma)^{\lambda_2} + (\zeta-1)(1-\gamma)^{\lambda_2}}\right)^{\lambda_1}}{\left(1 + (\zeta-1)\frac{(1+(\zeta-1)\gamma)^{\lambda_2} - (1-\gamma)^{\lambda_2}}{(1+(\zeta-1)\gamma)^{\lambda_2} + (\zeta-1)(1-\gamma)^{\lambda_2}}\right)^{\lambda_1} + (\zeta-1)\left(1 - \frac{(1+(\zeta-1)\gamma)^{\lambda_2} - (1-\gamma)^{\lambda_2}}{(1+(\zeta-1)\gamma)^{\lambda_2} + (\zeta-1)(1-\gamma)^{\lambda_2}}\right)^{\lambda_1}} \right\}$

$= \cup_{\gamma \in h} \left\{ \frac{(1 + (\zeta-1)\gamma)^{(\lambda_1 \lambda_2)} - (1-\gamma)^{(\lambda_1 \lambda_2)}}{(1 + (\zeta-1)\gamma)^{(\lambda_1 \lambda_2)} + (\zeta-1)(1-\gamma)^{(\lambda_1 \lambda_2)}} \right\}$

$= (\lambda_1 \lambda_2) \cdot_H h$.

(4) Since $h_1 \oplus_H h_2 = \cup_{\gamma_1 \in h_1, \gamma_2 \in h_2} \left\{ \frac{\gamma_1 + \gamma_2 - \gamma_1 \gamma_2 - (1-\zeta)\gamma_1 \gamma_2}{1 - (1-\zeta)\gamma_1 \gamma_2} \right\}$, by the operational law (3) in Definition 4, we have

$$\lambda \cdot_H (h_1 \oplus_H h_2)$$

$$= \cup_{\gamma_1 \in h_1, \gamma_2 \in h_2} \left\{ \frac{(1 + (\zeta - 1)\frac{\gamma_1 + \gamma_2 - \gamma_1 \gamma_2 - (1-\zeta)\gamma_1 \gamma_2}{1 - (1-\zeta)\gamma_1 \gamma_2})^\lambda - (1 - \frac{\gamma_1 + \gamma_2 - \gamma_1 \gamma_2 - (1-\zeta)\gamma_1 \gamma_2}{1 - (1-\zeta)\gamma_1 \gamma_2})^\lambda}{(1 + (\zeta - 1)\frac{\gamma_1 + \gamma_2 - \gamma_1 \gamma_2 - (1-\zeta)\gamma_1 \gamma_2}{1 - (1-\zeta)\gamma_1 \gamma_2})^\lambda + (\zeta - 1)(1 - \frac{\gamma_1 + \gamma_2 - \gamma_1 \gamma_2 - (1-\zeta)\gamma_1 \gamma_2}{1 - (1-\zeta)\gamma_1 \gamma_2})^\lambda} \right\}$$

$$= \cup_{\gamma_1 \in h_1, \gamma_2 \in h_2} \left\{ \frac{((1 + (\zeta - 1)\gamma_1)(1 + (\zeta - 1)\gamma_2))^\lambda - ((1 - \gamma_1)(1 - \gamma_2))^\lambda}{((1 + (\zeta - 1)\gamma_1)(1 + (\zeta - 1)\gamma_2))^\lambda + (\zeta - 1)((1 - \gamma_1)(1 - \gamma_2))^\lambda} \right\}.$$

Since $\lambda \cdot_H h_1 = \cup_{\gamma_1 \in h_1} \left\{ \frac{(1+(\zeta-1)\gamma_1)^\lambda - (1-\gamma_1)^\lambda}{(1+(\zeta-1)\gamma_1)^\lambda + (\zeta-1)(1-\gamma_1)^\lambda} \right\}$ and $\lambda \cdot_H h_2 = \cup_{\gamma_2 \in h_2} \left\{ \frac{(1+(\zeta-1)\gamma_2)^\lambda - (1-\gamma_2)^\lambda}{(1+(\zeta-1)\gamma_2)^\lambda + (\zeta-1)(1-\gamma_2)^\lambda} \right\}$, we have

$$(\lambda \cdot_H h_1) \oplus_H (\lambda \cdot_H h_2)$$

$$= \cup_{\gamma_1 \in h_1, \gamma_2 \in h_2} \left\{ \frac{\left[ \begin{array}{c} \frac{(1+(\zeta-1)\gamma_1)^\lambda - (1-\gamma_1)^\lambda}{(1+(\zeta-1)\gamma_1)^\lambda + (\zeta-1)(1-\gamma_1)^\lambda} + \frac{(1+(\zeta-1)\gamma_2)^\lambda - (1-\gamma_2)^\lambda}{(1+(\zeta-1)\gamma_2)^\lambda + (\zeta-1)(1-\gamma_2)^\lambda} \\ - \frac{(1+(\zeta-1)\gamma_1)^\lambda - (1-\gamma_1)^\lambda}{(1+(\zeta-1)\gamma_1)^\lambda + (\zeta-1)(1-\gamma_1)^\lambda} \frac{(1+(\zeta-1)\gamma_2)^\lambda - (1-\gamma_2)^\lambda}{(1+(\zeta-1)\gamma_2)^\lambda + (\zeta-1)(1-\gamma_2)^\lambda} \\ -(1-\zeta)\frac{(1+(\zeta-1)\gamma_1)^\lambda - (1-\gamma_1)^\lambda}{(1+(\zeta-1)\gamma_1)^\lambda + (\zeta-1)(1-\gamma_1)^\lambda} \cdot \frac{(1+(\zeta-1)\gamma_2)^\lambda - (1-\gamma_2)^\lambda}{(1+(\zeta-1)\gamma_2)^\lambda + (\zeta-1)(1-\gamma_2)^\lambda} \end{array} \right]}{1 - (1-\zeta)\frac{(1+(\zeta-1)\gamma_1)^\lambda - (1-\gamma_1)^\lambda}{(1+(\zeta-1)\gamma_1)^\lambda + (\zeta-1)(1-\gamma_1)^\lambda} \cdot \frac{(1+(\zeta-1)\gamma_2)^\lambda - (1-\gamma_2)^\lambda}{(1+(\zeta-1)\gamma_2)^\lambda + (\zeta-1)(1-\gamma_2)^\lambda}} \right\}$$

$$= \cup_{\gamma_1 \in h_1, \gamma_2 \in h_2} \left\{ \frac{((1 + (\zeta - 1)\gamma_1)(1 + (\zeta - 1)\gamma_2))^\lambda - ((1 - \gamma_1)(1 - \gamma_2))^\lambda}{((1 + (\zeta - 1)\gamma_1)(1 + (\zeta - 1)\gamma_2))^\lambda + (\zeta - 1)((1 - \gamma_1)(1 - \gamma_2))^\lambda} \right\}.$$

Hence $\lambda \cdot_H (h_1 \oplus_H h_2) = (\lambda \cdot_H h_1) \oplus_H (\lambda \cdot_H h_2)$.

(7) Since $h_1 \otimes_H h_2 = \cup_{\gamma_1 \in h_1, \gamma_2 \in h_2} \left\{ \frac{\gamma_1 \gamma_2}{\zeta + (1-\zeta)(\gamma_1 + \gamma_2 - \gamma_1 \gamma_2)} \right\}$, by the operational law (4) in Definition 4, we have

$$(h_1 \otimes_H h_2)^{\wedge_H \lambda}$$

$$= \cup_{\gamma_1 \in h_1, \gamma_2 \in h_2} \left\{ \frac{\zeta \left( \frac{\gamma_1 \gamma_2}{\zeta + (1-\zeta)(\gamma_1 + \gamma_2 - \gamma_1 \gamma_2)} \right)^\lambda}{\left( 1 + (\zeta - 1)(1 - \frac{\gamma_1 \gamma_2}{\zeta + (1-\zeta)(\gamma_1 + \gamma_2 - \gamma_1 \gamma_2)}) \right)^\lambda + (\zeta - 1) \left( \frac{\gamma_1 \gamma_2}{\zeta + (1-\zeta)(\gamma_1 + \gamma_2 - \gamma_1 \gamma_2)} \right)^\lambda} \right\}$$

$$= \cup_{\gamma_1 \in h_1, \gamma_2 \in h_2} \left\{ \frac{\zeta \gamma_1^\lambda \gamma_2^\lambda}{((1 + (\zeta - 1)(1 - \gamma_1))(1 + (\zeta - 1)(1 - \gamma_2)))^\lambda + (\zeta - 1)\gamma_1^\lambda \gamma_2^\lambda} \right\}.$$

Since $h_1^{\wedge_H \lambda} = \cup_{\gamma_1 \in h_1} \left\{ \frac{\zeta \gamma_1^\lambda}{(1+(\zeta-1)(1-\gamma_1))^\lambda + (\zeta-1)\gamma_1^\lambda} \right\}$ and $h_2^{\wedge_H \lambda} = \cup_{\gamma_2 \in h_2} \left\{ \frac{\zeta \gamma_2^\lambda}{(1+(\zeta-1)(1-\gamma_2))^\lambda + (\zeta-1)\gamma_2^\lambda} \right\}$,
we have

$$h_1^{\wedge_H \lambda} \otimes_H h_2^{\wedge_H \lambda}$$

$$= \cup_{\gamma_1 \in h_1, \gamma_2 \in h_2} \left\{ \frac{\frac{\zeta \gamma_1^\lambda}{(1+(\zeta-1)(1-\gamma_1))^\lambda + (\zeta-1)\gamma_1^\lambda} \cdot \frac{\zeta \gamma_2^\lambda}{(1+(\zeta-1)(1-\gamma_2))^\lambda + (\zeta-1)\gamma_2^\lambda}}{\left[ \zeta + (1-\zeta)\left( \frac{\zeta \gamma_1^\lambda}{(1+(\zeta-1)(1-\gamma_1))^\lambda + (\zeta-1)\gamma_1^\lambda} + \frac{\zeta \gamma_2^\lambda}{(1+(\zeta-1)(1-\gamma_2))^\lambda + (\zeta-1)\gamma_2^\lambda} - \frac{\zeta \gamma_1^\lambda}{(1+(\zeta-1)(1-\gamma_1))^\lambda + (\zeta-1)\gamma_1^\lambda} \cdot \frac{\zeta \gamma_2^\lambda}{(1+(\zeta-1)(1-\gamma_2))^\lambda + (\zeta-1)\gamma_2^\lambda} \right) \right]} \right\}$$

$$= \cup_{\gamma_1 \in h_1, \gamma_2 \in h_2} \left\{ \frac{\zeta \gamma_1^\lambda \gamma_2^\lambda}{((1+(\zeta-1)(1-\gamma_1))(1+(\zeta-1)(1-\gamma_2)))^\lambda + (\zeta-1)\gamma_1^\lambda \gamma_2^\lambda} \right\}.$$

Hence $(h_1 \otimes_H h_2)^{\wedge_H \lambda} = h_1^{\wedge_H \lambda} \otimes_H h_2^{\wedge_H \lambda}$.

(8) Since $h^{\wedge_H \lambda_1} = \cup_{\gamma \in h} \left\{ \frac{\zeta \gamma^{\lambda_1}}{(1+(\zeta-1)(1-\gamma))^{\lambda_1} + (\zeta-1)\gamma^{\lambda_1}} \right\}$, we have

$$(h^{\wedge_H \lambda_1})^{\wedge_H \lambda_2}$$

$$= \cup_{\gamma \in h} \left\{ \frac{\zeta \left( \frac{\zeta \gamma^{\lambda_1}}{(1+(\zeta-1)(1-\gamma))^{\lambda_1} + (\zeta-1)\gamma^{\lambda_1}} \right)^{\lambda_2}}{\left( 1 + (\zeta-1)(1 - \frac{\zeta \gamma^{\lambda_1}}{(1+(\zeta-1)(1-\gamma))^{\lambda_1} + (\zeta-1)\gamma^{\lambda_1}}) \right)^{\lambda_2} + (\zeta-1)\left( \frac{\zeta \gamma^{\lambda_1}}{(1+(\zeta-1)(1-\gamma))^{\lambda_1} + (\zeta-1)\gamma^{\lambda_1}} \right)^{\lambda_2}} \right\}$$

$$= \cup_{\gamma \in h} \left\{ \frac{\zeta \gamma^{(\lambda_1 \lambda_2)}}{(1+(\zeta-1)(1-\gamma))^{(\lambda_1 \lambda_2)} + (\zeta-1)\gamma^{(\lambda_1 \lambda_2)}} \right\}$$

$$= h^{\wedge_H (\lambda_1 \lambda_2)}.$$

□

However, the operational laws $(\lambda_1 \cdot_H h) \oplus_H (\lambda_2 \cdot_H h) = (\lambda_1 + \lambda_2) \cdot_H h$ and $h^{\wedge_H \lambda_1} \otimes_H h^{\wedge_H \lambda_2} = h^{\wedge_H (\lambda_1 + \lambda_2)}$ do not hold in general. To illustrate these, we give an example as follows:

**Example 1.** *Let $h = \{0.3, 0.5\}$, $\lambda_1 = \lambda_2 = 1$ and $\zeta = 3$, then*

$$(\lambda_1 \cdot_H h) \oplus_H (\lambda_2 \cdot_H h) = h \oplus_H h = \cup_{i,j=1,2} \left\{ \frac{\gamma_i + \gamma_j + \gamma_i \gamma_j}{1 + 2\gamma_i \gamma_j} \right\}$$
$$= \{0.5874, 0.7308, 0.7308, 0.8333\},$$
$$(\lambda_1 + \lambda_2) \cdot_H h = 2 \cdot_H h = \cup_{i=1,2} \left\{ \frac{(1+2\gamma_i)^2 - (1-\gamma_i)^2}{(1+2\gamma_i)^2 + 2(1-\gamma_i)^2} \right\} = \{0.5874, 0.8333\}.$$

*From Definition 3, we have $s((\lambda_1 \cdot_H h) \oplus_H (\lambda_2 \cdot_H h)) = 0.7199 > 0.7104 = s((\lambda_1 + \lambda_2) \cdot_H h)$ and thus $(\lambda_1 \cdot_H h) \oplus_H (\lambda_2 \cdot_H h) \neq (\lambda_1 + \lambda_2) \cdot_H h$. Furthermore, we have $s(h^{\wedge_H \lambda_1} \otimes_H h^{\wedge_H \lambda_2}) = 0.0971 < 0.1060 = s(h^{\wedge_H (\lambda_1 + \lambda_2)})$ and thus $h^{\wedge_H \lambda_1} \otimes_H h^{\wedge_H \lambda_2} \neq h^{\wedge_H (\lambda_1 + \lambda_2)}$.*

## 3. Hesitant Fuzzy Hamacher Power-Weighted Aggregation Operators

In this section, based on the Hamacher operation, we shall extend the power-aggregation operators to accommodate the situations where the input arguments are HFEs.

*3.1. Hesitant Fuzzy Hamacher Power-Weighted Average/geometric Operators*

Based on the PA operator [24] and hesitant fuzzy Hamacher weighted average (HFHWA) operator [17], we firstly define the hesitant fuzzy Hamacher power-weighted average (HFHPWA) operator as follows.

**Definition 5.** *Let $h_i$ $(i = 1, 2, \ldots, n)$ be a collection of HFEs and $w = (w_1, w_2, \ldots, w_n)^T$ be the weight vector of $h_i$ $(i = 1, 2, \ldots, n)$ such that $w_i \in [0, 1]$ and $\sum_{i=1}^{n} w_i = 1$. A hesitant fuzzy Hamacher power-weighted average (HFHPWA) operator is a function $H^n \to H$ such that*

$$\text{HFHPWA}_\zeta(h_1, h_2, \ldots, h_n) = \oplus_H{}_{i=1}^n \left( \frac{w_i(1 + T(h_i)) \cdot_H h_i}{\sum_{i=1}^n w_i(1 + T(h_i))} \right), \tag{5}$$

*where parameter $\zeta > 0$, $T(h_i) = \sum_{j=1, j \neq i}^n w_j \text{Sup}(h_i, h_j)$ and $\text{Sup}(h_i, h_j)$ is the support for $h_i$ from $h_j$, satisfying the following conditions:*

*(1) $\text{Sup}(h_i, h_j) \in [0, 1]$;*
*(2) $\text{Sup}(h_i, h_j) = \text{Sup}(h_j, h_i)$;*
*(3) $\text{Sup}(h_i, h_j) \geq \text{Sup}(h_s, h_t)$ if $d(h_i, h_j) \leq d(h_s, h_t)$, where $d$ is the hesitant normalized Hamming distance measure between two HFEs given in Equation (4).*

Here, the support measure (Sup) can be used to measure the closeness of a preference value with other preference value because it is essentially similarity measure.

**Theorem 2.** *Let $h_i$ $(i = 1, 2, \ldots, n)$ be a collection of HFEs and $w = (w_1, w_2, \ldots, w_n)^T$ be the weight vector of $h_i$ $(i = 1, 2, \ldots, n)$ such that $w_i \in [0, 1]$ and $\sum_{i=1}^{n} w_i = 1$, then the aggregated value by HFHPWA operator is also a HFE, and*

$$\text{HFHPWA}_\zeta(h_1, h_2, \ldots, h_n)$$

$$= \cup_{\gamma_1 \in h_1, \gamma_2 \in h_2, \ldots, \gamma_n \in h_n} \left\{ \frac{\prod_{i=1}^n (1 + (\zeta - 1)\gamma_i)^{\frac{w_i(1+T(h_i))}{\sum_{i=1}^n w_i(1+T(h_i))}} - \prod_{i=1}^n (1 - \gamma_i)^{\frac{w_i(1+T(h_i))}{\sum_{i=1}^n w_i(1+T(h_i))}}}{\prod_{i=1}^n (1 + (\zeta - 1)\gamma_i)^{\frac{w_i(1+T(h_i))}{\sum_{i=1}^n w_i(1+T(h_i))}} + (\zeta - 1)\prod_{i=1}^n (1 - \gamma_i)^{\frac{w_i(1+T(h_i))}{\sum_{i=1}^n w_i(1+T(h_i))}}} \right\}. \tag{6}$$

**Proof.** Equation (6) can be proved by mathematical induction on $n$ as follows.

For $n = 1$, the result of Equation (6) is clear.

Suppose that Equation (6) holds for $n = k$, that is

$$\text{HFHPWA}_\zeta(h_1, h_2, \ldots, h_k)$$

$$= \cup_{\gamma_1 \in h_1, \gamma_2 \in h_2, \ldots, \gamma_k \in h_k} \left\{ \frac{\prod_{i=1}^k (1 + (\zeta - 1)\gamma_i)^{\frac{w_i(1+T(h_i))}{\sum_{i=1}^k w_i(1+T(h_i))}} - \prod_{i=1}^k (1 - \gamma_i)^{\frac{w_i(1+T(h_i))}{\sum_{i=1}^k w_i(1+T(h_i))}}}{\prod_{i=1}^k (1 + (\zeta - 1)\gamma_i)^{\frac{w_i(1+T(h_i))}{\sum_{i=1}^k w_i(1+T(h_i))}} + (\zeta - 1)\prod_{i=1}^k (1 - \gamma_i)^{\frac{w_i(1+T(h_i))}{\sum_{i=1}^k w_i(1+T(h_i))}}} \right\}.$$

Then, when $n = k + 1$, by Definitions 4 and 5, we have

$$\text{HFHPWA}_\zeta(h_1, h_2, \ldots, h_{k+1}) = \oplus_{H\,i=1}^{k}\left(\frac{w_i(1 + T(h_i)) \cdot_H h_i}{\sum_{i=1}^{k+1} w_i(1 + T(h_i))}\right) \oplus_H \left(\frac{w_{k+1}(1 + T(h_{k+1})) \cdot_H h_{k+1}}{\sum_{i=1}^{k+1} w_i(1 + T(h_i))}\right)$$

$$= \cup_{\gamma_1 \in h_1, \gamma_2 \in h_2, \ldots, \gamma_k \in h_k} \left\{\frac{\prod_{i=1}^{k}(1 + (\zeta - 1)\gamma_i)^{\frac{w_i(1+T(h_i))}{\sum_{i=1}^{k+1} w_i(1+T(h_i))}} - \prod_{i=1}^{k}(1 - \gamma_i)^{\frac{w_i(1+T(h_i))}{\sum_{i=1}^{k+1} w_i(1+T(h_i))}}}{\prod_{i=1}^{k}(1 + (\zeta - 1)\gamma_i)^{\frac{w_i(1+T(h_i))}{\sum_{i=1}^{k+1} w_i(1+T(h_i))}} + (\zeta - 1)\prod_{i=1}^{k}(1 - \gamma_i)^{\frac{w_i(1+T(h_i))}{\sum_{i=1}^{k+1} w_i(1+T(h_i))}}}\right\}$$

$$\oplus_H \cup_{\gamma_{k+1} \in h_{k+1}} \left\{\frac{(1 + (\zeta - 1)\gamma_{k+1})^{\frac{w_{k+1}(1+T(h_{k+1}))}{\sum_{i=1}^{k+1} w_i(1+T(h_i))}} - (1 - \gamma_{k+1})^{\frac{w_{k+1}(1+T(h_{k+1}))}{\sum_{i=1}^{k+1} w_i(1+T(h_i))}}}{(1 + (\zeta - 1)\gamma_{k+1})^{\frac{w_{k+1}(1+T(h_{k+1}))}{\sum_{i=1}^{k+1} w_i(1+T(h_i))}} + (\zeta - 1)(1 - \gamma_{k+1})^{\frac{w_{k+1}(1+T(h_{k+1}))}{\sum_{i=1}^{k+1} w_i(1+T(h_i))}}}\right\}.$$

Let $a_1 = \prod_{i=1}^{k}(1 + (\zeta - 1)\gamma_i)^{\frac{w_i(1+T(h_i))}{\sum_{i=1}^{k+1} w_i(1+T(h_i))}}$, $b_1 = \prod_{i=1}^{k}(1 - \gamma_i)^{\frac{w_i(1+T(h_i))}{\sum_{i=1}^{k+1} w_i(1+T(h_i))}}$, $a_2 = (1 + (\zeta - 1)\gamma_{k+1})^{\frac{w_{k+1}(1+T(h_{k+1}))}{\sum_{i=1}^{k+1} w_i(1+T(h_i))}}$ and $b_2 = (1 - \gamma_{k+1})^{\frac{w_{k+1}(1+T(h_{k+1}))}{\sum_{i=1}^{k+1} w_i(1+T(h_i))}}$, then

$$\text{HFHPWA}_\zeta(h_1, h_2, \ldots, h_{k+1})$$

$$= \cup_{\gamma_1 \in h_1, \gamma_2 \in h_2, \ldots, \gamma_k \in h_k}\left\{\frac{a_1 - b_1}{a_1 + (\zeta - 1)b_1}\right\} \oplus_H \cup_{\gamma_{k+1} \in h_{k+1}}\left\{\frac{a_2 - b_2}{a_2 + (\zeta - 1)b_2}\right\}$$

$$= \cup_{\gamma_1 \in h_1, \ldots, \gamma_k \in h_k, \gamma_{k+1} \in h_{k+1}}\left\{\frac{\left[\frac{a_1 - b_1}{a_1 + (\zeta - 1)b_1} + \frac{a_2 - b_2}{a_2 + (\zeta - 1)b_2} - \frac{a_1 - b_1}{a_1 + (\zeta - 1)b_1} \cdot \frac{a_2 - b_2}{a_2 + (\zeta - 1)b_2}\right] - (1 - \zeta)\frac{a_1 - b_1}{a_1 + (\zeta - 1)b_1} \cdot \frac{a_2 - b_2}{a_2 + (\zeta - 1)b_2}}{1 - (1 - \zeta)\frac{a_1 - b_1}{a_1 + (\zeta - 1)b_1} \cdot \frac{a_2 - b_2}{a_2 + (\zeta - 1)b_2}}\right\}$$

$$= \cup_{\gamma_1 \in h_1, \gamma_2 \in h_2, \ldots, \gamma_{k+1} \in h_{k+1}}\left\{\frac{a_1 a_2 - b_1 b_2}{a_1 a_2 + (\zeta - 1)b_1 b_2}\right\}$$

$$= \cup_{\gamma_1 \in h_1, \gamma_2 \in h_2, \ldots, \gamma_{k+1} \in h_{k+1}}\left\{\frac{\prod_{i=1}^{k+1}(1 + (\zeta - 1)\gamma_i)^{\frac{w_i(1+T(h_i))}{\sum_{i=1}^{k+1} w_i(1+T(h_i))}} - \prod_{i=1}^{k+1}(1 - \gamma_i)^{\frac{w_i(1+T(h_i))}{\sum_{i=1}^{k+1} w_i(1+T(h_i))}}}{\prod_{i=1}^{k+1}(1 + (\zeta - 1)\gamma_i)^{\frac{w_i(1+T(h_i))}{\sum_{i=1}^{k+1} w_i(1+T(h_i))}} + (\zeta - 1)\prod_{i=1}^{k+1}(1 - \gamma_i)^{\frac{w_i(1+T(h_i))}{\sum_{i=1}^{k+1} w_i(1+T(h_i))}}}\right\},$$

i.e., Equation (6) holds for $n = k + 1$. Thus, Equation (6) holds for all $n$. □

**Remark 1.** *(1) If* $\text{Sup}(h_i, h_j) = k$, *for all* $i \neq j$, *then*

$$\text{HFHPWA}_\zeta(h_1, h_2, \ldots, h_n) = \oplus_{H\,i=1}^{n}(w_i \cdot_H h_i)$$

$$= \cup_{\gamma_1 \in h_1, \gamma_2 \in h_2, \ldots, \gamma_n \in h_n}\left\{\frac{\prod_{i=1}^{n}(1 + (\zeta - 1)\gamma_i)^{w_i} - \prod_{i=1}^{n}(1 - \gamma_i)^{w_i}}{\prod_{i=1}^{n}(1 + (\zeta - 1)\gamma_i)^{w_i} + (\zeta - 1)\prod_{i=1}^{n}(1 - \gamma_i)^{w_i}}\right\}, \quad (7)$$

*which indicates that when all supports are the same, the HFHPWA operator reduces to the hesitant fuzzy Hamacher weighted average (HFHWA) operator [17].*

*(2) For the HFHPWA operator, if* $\zeta = 1$, *then the HFHPWA operator reduces to the following:*

$$\text{HFHPWA}_1(h_1, h_2, \ldots, h_n) = \cup_{\gamma_1 \in h_1, \gamma_2 \in h_2, \ldots, \gamma_n \in h_n}\left\{1 - \prod_{i=1}^{n}(1 - \gamma_i)^{\frac{w_i(1+T(h_i))}{\sum_{i=1}^{n} w_i(1+T(h_i))}}\right\} \quad (8)$$

*which is called the hesitant fuzzy power-weighted average (HFPWA) operator and if $\zeta = 2$, then the HFHPWA operator reduces to the hesitant fuzzy Einstein power-weighted average (HFEPWA) operator [18]:*

$$
\begin{aligned}
&\text{HFHPWA}_2(h_1, h_2, \ldots, h_n) \\
&= \cup_{\gamma_1 \in h_1, \gamma_2 \in h_2, \ldots, \gamma_n \in h_n} \left\{ \frac{\prod_{i=1}^n (1 + \gamma_i)^{\frac{w_i(1+T(h_i))}{\sum_{i=1}^n w_i(1+T(h_i))}} - \prod_{i=1}^n (1 - \gamma_i)^{\frac{w_i(1+T(h_i))}{\sum_{i=1}^n w_i(1+T(h_i))}}}{\prod_{i=1}^n (1 + \gamma_i)^{\frac{w_i(1+T(h_i))}{\sum_{i=1}^n w_i(1+T(h_i))}} + \prod_{i=1}^n (1 - \gamma_i)^{\frac{w_i(1+T(h_i))}{\sum_{i=1}^n w_i(1+T(h_i))}}} \right\}. \quad (9)
\end{aligned}
$$

To analyze the relationship between the HFHPWA operator and the HFPWA operator, we introduce the following lemma.

**Lemma 1.** *[47,48] Let $x_i > 0$, $w_i > 0$, $i = 1, 2, \ldots, n$, and $\sum_{i=1}^n w_i = 1$, then $\prod_{i=1}^n x_i^{w_i} \leq \sum_{i=1}^n w_i x_i$, with equality if and only if $x_1 = x_2 = \cdots = x_n$.*

**Theorem 3.** *Let $h_i$ ($i = 1, 2, \ldots, n$) be a collection of HFEs and $w = (w_1, w_2, \ldots, w_n)^T$ be the weight vector of $h_i$ such that $w_i \in [0, 1]$ and $\sum_{i=1}^n w_i = 1$, then*

$$
\text{HFHPWA}_\zeta(h_1, h_2, \ldots, h_n) \leq \text{HFPWA}(h_1, h_2, \ldots, h_n).
$$

**Proof.** For any $\gamma_i \in h_i$ ($i = 1, 2, \ldots, n$), by Lemma 1, we have

$$
\begin{aligned}
&\prod_{i=1}^n (1 + (\zeta - 1)\gamma_i)^{\frac{w_i(1+T(h_i))}{\sum_{i=1}^n w_i(1+T(h_i))}} + (\zeta - 1) \prod_{i=1}^n (1 - \gamma_i)^{\frac{w_i(1+T(h_i))}{\sum_{i=1}^n w_i(1+T(h_i))}} \\
&\leq \sum_{i=1}^n \frac{w_i(1+T(h_i))}{\sum_{i=1}^n w_i(1+T(h_i))} (1 + (\zeta - 1)\gamma_i) + (\zeta - 1) \sum_{i=1}^n \frac{w_i(1+T(h_i))}{\sum_{i=1}^n w_i(1+T(h_i))} (1 - \gamma_i) = \zeta.
\end{aligned}
$$

Then,

$$
\begin{aligned}
&\frac{\prod_{i=1}^n (1 + (\zeta - 1)\gamma_i)^{\frac{w_i(1+T(h_i))}{\sum_{i=1}^n w_i(1+T(h_i))}} - \prod_{i=1}^n (1 - \gamma_i)^{\frac{w_i(1+T(h_i))}{\sum_{i=1}^n w_i(1+T(h_i))}}}{\prod_{i=1}^n (1 + (\zeta - 1)\gamma_i)^{\frac{w_i(1+T(h_i))}{\sum_{i=1}^n w_i(1+T(h_i))}} + (\zeta - 1) \prod_{i=1}^n (1 - \gamma_i)^{\frac{w_i(1+T(h_i))}{\sum_{i=1}^n w_i(1+T(h_i))}}} \\
&= 1 - \frac{\zeta \prod_{i=1}^n (1 - \gamma_i)^{\frac{w_i(1+T(h_i))}{\sum_{i=1}^n w_i(1+T(h_i))}}}{\prod_{i=1}^n (1 + (\zeta - 1)\gamma_i)^{\frac{w_i(1+T(h_i))}{\sum_{i=1}^n w_i(1+T(h_i))}} + (\zeta - 1) \prod_{i=1}^n (1 - \gamma_i)^{\frac{w_i(1+T(h_i))}{\sum_{i=1}^n w_i(1+T(h_i))}}} \\
&\leq 1 - \frac{\zeta \prod_{i=1}^n (1 - \gamma_i)^{\frac{w_i(1+T(h_i))}{\sum_{i=1}^n w_i(1+T(h_i))}}}{\zeta} = 1 - \prod_{i=1}^n (1 - \gamma_i)^{\frac{w_i(1+T(h_i))}{\sum_{i=1}^n w_i(1+T(h_i))}},
\end{aligned}
$$

which implies that $\oplus_{H_{i=1}^n} \left( \frac{w_i(1+T(h_i)) \cdot_H h_i}{\sum_{i=1}^n w_i(1+T(h_i))} \right) \leq \oplus_{i=1}^n \left( \frac{w_i(1+T(h_i))h_i}{\sum_{i=1}^n w_i(1+T(h_i))} \right)$. Thus, we obtain $\text{HFHPWA}_\zeta(h_1, h_2, \ldots, h_n) \leq \text{HFPWA}(h_1, h_2, \ldots, h_n)$. $\square$

Theorem 3 shows that the values aggregated by the HFHPWA operator are not larger than those obtained by the HFPWA operator. That is to say, the HFHPWA operator reflects the decision-maker's pessimistic attitude than the HFPWA operator in aggregation process. Furthermore, based on Theorem 2, we have the properties of the HFHPWA operator as follows.

**Theorem 4.** *Let $h_i$ ($i = 1, 2, \ldots, n$) be a collection of HFEs and $w = (w_1, w_2, \ldots, w_n)^T$ be the weight vector of $h_i$ such that $w_i \in [0, 1]$ and $\sum_{i=1}^n w_i = 1$, then we have the followings:*
*(1) Boundedness: If $h^- = \min\{\gamma_i | \gamma_i \in h_i\}$ and $h^+ = \max\{\gamma_i | \gamma_i \in h_i\}$, then*

$$
h^- \leq \text{HFHPWA}_\zeta(h_1, h_2, \ldots, h_n) \leq h^+.
$$

(2) *Monotonicity: Let $h'_i$ ($i = 1, 2, \ldots, n$) be a collection of HFEs, if $w = (w_1, w_2, \ldots, w_n)^T$ is also the weight vector of $h'_i$, and $\gamma_i \leq \gamma'_i$ for any $h_i$ and $h'_i$ ($i = 1, 2, \ldots, n$), then*

$$\text{HFHPWA}_\zeta(h_1, h_2, \ldots, h_n) \leq \text{HFHPWA}_\zeta(h'_1, h'_2, \ldots, h'_n).$$

**Proof.** (1) Let $f(x) = \frac{1+(\zeta-1)x}{1-x}$, $x \in [0,1)$, then $f'(x) = \frac{\zeta}{(1-x)^2} > 0$ and thus $f(x)$ is an increasing function. Since $h^- \leq \gamma_i \leq h^+$ for all $i$, then $f(h^-) \leq f(\gamma_i) \leq f(h^+)$, i.e., $\frac{1+(\zeta-1)h^-}{1-h^-} \leq \frac{1+(\zeta-1)\gamma_i}{1-\gamma_i} \leq \frac{1+(\zeta-1)h^+}{1-h^+}$. For convenience, let $t_i = \frac{w_i(1+T(h_i))}{\sum_{i=1}^n w_i(1+T(h_i))}$. Since $w = (w_1, w_2, \ldots, w_n)^T$ is the weight vector of $h_i$ satisfying $w_i \in [0,1]$ and $\sum_{i=1}^n w_i = 1$, then for all $i$, we have

$$\left(\frac{1+(\zeta-1)h^-}{1-h^-}\right)^{t_i} \leq \left(\frac{1+(\zeta-1)\gamma_i}{1-\gamma_i}\right)^{t_i} \leq \left(\frac{1+(\zeta-1)h^+}{1-h^+}\right)^{t_i}$$

$$\Leftrightarrow \left(1+\frac{\zeta h^-}{1-h^-}\right)^{t_i} \leq \left(\frac{1+(\zeta-1)\gamma_i}{1-\gamma_i}\right)^{t_i} \leq \left(1+\frac{\zeta h^+}{1-h^+}\right)^{t_i}$$

$$\Leftrightarrow \prod_{i=1}^n \left(1+\frac{\zeta h^-}{1-h^-}\right)^{t_i} \leq \prod_{i=1}^n \left(\frac{1+(\zeta-1)\gamma_i}{1-\gamma_i}\right)^{t_i} \leq \prod_{i=1}^n \left(1+\frac{\zeta h^+}{1-h^+}\right)^{t_i}$$

$$\Leftrightarrow 1+\frac{\zeta h^-}{1-h^-} \leq \prod_{i=1}^n \left(\frac{1+(\zeta-1)\gamma_i}{1-\gamma_i}\right)^{t_i} \leq 1+\frac{\zeta h^+}{1-h^+}$$

$$\Leftrightarrow \zeta+\frac{\zeta h^-}{1-h^-} \leq \prod_{i=1}^n \left(\frac{1+(\zeta-1)\gamma_i}{1-\gamma_i}\right)^{t_i} + (\zeta-1) \leq \zeta+\frac{\zeta h^+}{1-h^+}$$

$$\Leftrightarrow \frac{1}{\zeta+\frac{\zeta h^+}{1-h^+}} \leq \frac{1}{\prod_{i=1}^n \left(\frac{1+(\zeta-1)\gamma_i}{1-\gamma_i}\right)^{t_i} + (\zeta-1)} \leq \frac{1}{\zeta+\frac{\zeta h^-}{1-h^-}}$$

$$\Leftrightarrow \frac{1-h^+}{\zeta} \leq \frac{\prod_{i=1}^n (1-\gamma_i)^{t_i}}{\prod_{i=1}^n (1+(\zeta-1)\gamma_i)^{t_i} + (\zeta-1)\prod_{i=1}^n (1-\gamma_i)^{t_i}} \leq \frac{1-h^-}{\zeta}$$

$$\Leftrightarrow 1-h^+ \leq \frac{\zeta\prod_{i=1}^n (1-\gamma_i)^{t_i}}{\prod_{i=1}^n (1+(\zeta-1)\gamma_i)^{t_i} + (\zeta-1)\prod_{i=1}^n (1-\gamma_i)^{t_i}} \leq 1-h^-$$

$$\Leftrightarrow h^- \leq 1-\frac{\zeta\prod_{i=1}^n (1-\gamma_i)^{t_i}}{\prod_{i=1}^n (1+(\zeta-1)\gamma_i)^{t_i} + (\zeta-1)\prod_{i=1}^n (1-\gamma_i)^{t_i}} \leq h^+$$

$$\Leftrightarrow h^- \leq \frac{\prod_{i=1}^n (1+(\zeta-1)\gamma_i)^{t_i} - \prod_{i=1}^n (1-\gamma_i)^{t_i}}{\prod_{i=1}^n (1+(\zeta-1)\gamma_i)^{t_i} + (\zeta-1)\prod_{i=1}^n (1-\gamma_i)^{t_i}} \leq h^+.$$

Thus, we have $h^- \leq \text{HFHPWA}_\zeta(h_1, h_2, \ldots, h_n) \leq h^+$.

(2) Let $f(x) = \frac{1+(\zeta-1)x}{1-x}$, $x \in [0,1)$, then by (1), $f(x)$ is an increasing function. If for all $h_i$ and $h_i'$, $\gamma_i \leq \gamma_i'$, then $\frac{1+(\zeta-1)\gamma_i}{1-\gamma_i} \leq \frac{1+(\zeta-1)\gamma_i'}{1-\gamma_i'}$. For convenience, let $t_i = \frac{w_i(1+T(h_i))}{\sum_{i=1}^n w_i(1+T(h_i))}$, then we have

$$
\left(\frac{1+(\zeta-1)\gamma_i}{1-\gamma_i}\right)^{t_i} \leq \left(\frac{1+(\zeta-1)\gamma_i'}{1-\gamma_i'}\right)^{t_i}
$$

$$
\Leftrightarrow \prod_{i=1}^n \left(\frac{1+(\zeta-1)\gamma_i}{1-\gamma_i}\right)^{t_i} + (\zeta-1) \leq \prod_{i=1}^n \left(\frac{1+(\zeta-1)\gamma_i'}{1-\gamma_i'}\right)^{t_i} + (\zeta-1)
$$

$$
\Leftrightarrow \frac{1}{\prod_{i=1}^n \left(\frac{1+(\zeta-1)\gamma_i}{1-\gamma_i}\right)^{t_i} + (\zeta-1)} \geq \frac{1}{\prod_{i=1}^n \left(\frac{1+(\zeta-1)\gamma_i'}{1-\gamma_i'}\right)^{t_i} + (\zeta-1)}
$$

$$
\Leftrightarrow \frac{\zeta \prod_{i=1}^n (1-\gamma_i)^{t_i}}{\prod_{i=1}^n (1+(\zeta-1)\gamma_i)^{t_i} + (\zeta-1)\prod_{i=1}^n (1-\gamma_i)^{t_i}} \geq \frac{\zeta \prod_{i=1}^n (1-\gamma_i')^{t_i}}{\prod_{i=1}^n (1+(\zeta-1)\gamma_i')^{t_i} + (\zeta-1)\prod_{i=1}^n (1-\gamma_i')^{t_i}}
$$

$$
\Leftrightarrow 1 - \frac{\zeta \prod_{i=1}^n (1-\gamma_i)^{t_i}}{\left[\begin{array}{c}\prod_{i=1}^n (1+(\zeta-1)\gamma_i)^{t_i} \\ +(\zeta-1)\prod_{i=1}^n (1-\gamma_i)^{t_i}\end{array}\right]} \leq 1 - \frac{\zeta \prod_{i=1}^n (1-\gamma_i')^{t_i}}{\left[\begin{array}{c}\prod_{i=1}^n (1+(\zeta-1)\gamma_i')^{t_i} \\ +(\zeta-1)\prod_{i=1}^n (1-\gamma_i')^{t_i}\end{array}\right]}
$$

$$
\Leftrightarrow \frac{\prod_{i=1}^n (1+(\zeta-1)\gamma_i)^{t_i} - \prod_{i=1}^n (1-\gamma_i)^{t_i}}{\left[\begin{array}{c}\prod_{i=1}^n (1+(\zeta-1)\gamma_i)^{t_i} \\ +(\zeta-1)\prod_{i=1}^n (1-\gamma_i)^{t_i}\end{array}\right]} \leq \frac{\prod_{i=1}^n (1+(\zeta-1)\gamma_i')^{t_i} - \prod_{i=1}^n (1-\gamma_i')^{t_i}}{\left[\begin{array}{c}\prod_{i=1}^n (1+(\zeta-1)\gamma_i')^{t_i} \\ +(\zeta-1)\prod_{i=1}^n (1-\gamma_i')^{t_i}\end{array}\right]}.
$$

Thus, by Theorem 2, $\text{HFHPWA}_\zeta(h_1, h_2, \ldots, h_n) \leq \text{HFHPWA}_\zeta(h_1', h_2', \ldots, h_n')$. □

However, the HFHPWA operator is neither idempotent nor commutative, as illustrated by the following example.

**Example 2.** *Let $h_1 = \{0.8, 0.6\}$, $h_2 = \{0.9, 0.5\}$ and $h_3 = \{0.7, 0.6\}$ be three HFEs, $w = (0.3, 0.5, 0.2)^T$ be the weight vector of $h_1$, $h_2$ and $h_3$. Assume that $\text{Sup}(h_i, h_j)$ $(i, j = 1, 2, 3, i \neq j)$ is the support for $h_i$ from $h_j$ given by $\text{Sup}(h_i, h_j) = 1 - d(h_i, h_j)$, where $d(h_i, h_j)$ is the hesitant Hamming distance between $h_i$ and $h_j$. Then by Theorem 2, we have*

$$\text{HFHPWA}_5(h_1, h_2, h_3) = \{0.8388, 0.6555, 0.7942, 0.5797, 0.8261, 0.6332, 0.7786, 0.5549\},$$

$$\text{HFHPWA}_5(h_2, h_3, h_1) = \{0, 8013, 0.7683, 0.6723, 0.6255, 0.7650, 0.7275, 0.6208, 0.5701\},$$

$$\text{HFHPWA}_5(h_3, h_3, h_3) = \{0.7000, 0.6559, 0.6704, 0.6237, 0.6793, 0.6334, 0.6485, 0.6000\}.$$

*From Definition 3, we have $s(h_1) = s(h_2) = 0, 7$, $s(h_3) = 0.65$, $s(\text{HFHPWA}_5(h_1, h_2, h_3)) = 0.7076$, $s(\text{HFHPWA}_5(h_2, h_3, h_1)) = 0.6938$ and $s(\text{HFHPWA}_5(h_3, h_3, h_3)) = 0.6514$. Then $s(\text{HFHPWA}_5(h_3, h_3, h_3)) \neq s(h_3)$ and thus $\text{HFHPWA}_5(h_3, h_3, h_3) \neq h_3$, which implies that the HFHPWA operator is not idempotent. Furthermore, since $s(\text{HFHPWA}_5(h_1, h_2, h_3)) \neq s(\text{HFHPWA}_5(h_2, h_3, h_1))$, we have $\text{HFHPWA}_5(h_1, h_2, h_3) \neq \text{HFHPWA}_5(h_2, h_3, h_1)$. Thus, the HFHPWA operator is not commutative.*

Based on the power geometric (PG) operator [25] and hesitant fuzzy Hamacher weighted geometric (HFHWG) operator [17], we also define the hesitant fuzzy Hamacher power-weighted geometric operator as follows.

**Definition 6.** *Let $h_i$ $(i = 1, 2, \ldots, n)$ be a collection of HFEs and $w = (w_1, w_2, \ldots, w_n)^T$ be the weight vector of $h_i$ $(i = 1, 2, \ldots, n)$ such that $w_i \in [0, 1]$ and $\sum_{i=1}^n w_i = 1$. A hesitant fuzzy Hamacher power-weighted geometric (HFHPWG) operator is a function $H^n \to H$ such that*

$$
\text{HFHPWG}_\zeta(h_1, h_2, \ldots, h_n) = \otimes_{H i=1}^n \left(h_i^{\wedge_H \frac{w_i(1+T(h_i))}{\sum_{i=1}^n w_i(1+T(h_i))}}\right), \tag{10}
$$

*where parameter $\zeta > 0$, $T(h_i) = \sum_{j=1, j \neq i}^{n} w_j \text{Sup}(h_i, h_j)$ and $\text{Sup}(h_i, h_j)$ is the support for $h_i$ from $h_j$.*

**Theorem 5.** *Let $h_i$ $(i = 1, 2, \ldots, n)$ be a collection of HFEs and $w = (w_1, w_2, \ldots, w_n)^T$ be the weight vector of $h_i$ $(i = 1, 2, \ldots, n)$ such that $w_i \in [0, 1]$ and $\sum_{i=1}^{n} w_i = 1$, then the aggregated value by HFHPWG operator is also a HFE, and*

$$\text{HFHPWG}_\zeta(h_1, h_2, \ldots, h_n) = \cup_{\gamma_1 \in h_1, \gamma_2 \in h_2, \ldots, \gamma_n \in h_n} \left\{ \frac{\zeta \prod_{i=1}^{n} (\gamma_i)^{\frac{w_i(1+T(h_i))}{\sum_{i=1}^{n} w_i(1+T(h_i))}}}{\left[ \begin{array}{c} \prod_{i=1}^{n} (1 + (\zeta - 1)(1 - \gamma_i))^{\frac{w_i(1+T(h_i))}{\sum_{i=1}^{n} w_i(1+T(h_i))}} \\ +(\zeta - 1) \prod_{i=1}^{n} (\gamma_i)^{\frac{w_i(1+T(h_i))}{\sum_{i=1}^{n} w_i(1+T(h_i))}} \end{array} \right]} \right\}. \tag{11}$$

**Proof.** Similar to the proof of Theorem 2, Equation (11) can be proved by mathematical induction on $n$. $\square$

**Remark 2.** *(1) If $\text{Sup}(h_i, h_j) = k$, for all $i \neq j$, then*

$$\text{HFHPWG}_\zeta(h_1, h_2, \ldots, h_n) = \otimes_{H_{i=1}}^{n} \left( h_i^{\wedge_H w_i} \right) \tag{12}$$

*which indicates that when all supports are the same, the HFHPWG operator reduces to the hesitant fuzzy Hamacher weighted geometric (HFHWG) operator [17].*

*(2) If $\zeta = 1$, then then the HFHPWG operator reduces to the following:*

$$\text{HFHPWG}_1(h_1, h_2, \ldots, h_n) = \cup_{\gamma_1 \in h_1, \gamma_2 \in h_2, \ldots, \gamma_n \in h_n} \left\{ \prod_{i=1}^{n} (\gamma_i)^{\frac{w_i(1+T(h_i))}{\sum_{i=1}^{n} w_i(1+T(h_i))}} \right\} \tag{13}$$

*which is called the hesitant fuzzy power-weighted geometric (HFPWG) operator and if $\zeta = 2$, then the HFHPWG operator reduces to the hesitant fuzzy Einstein power-weighted geometric (HFEPWG) operator [18]:*

$$\text{HFHPWG}_2(h_1, h_2, \ldots, h_n)$$
$$= \cup_{\gamma_1 \in h_1, \gamma_2 \in h_2, \ldots, \gamma_n \in h_n} \left\{ \frac{2 \prod_{i=1}^{n} (\gamma_i)^{\frac{w_i(1+T(h_i))}{\sum_{i=1}^{n} w_i(1+T(h_i))}}}{\prod_{i=1}^{n} (2 - \gamma_i)^{\frac{w_i(1+T(h_i))}{\sum_{i=1}^{n} w_i(1+T(h_i))}} + \prod_{i=1}^{n} (\gamma_i)^{\frac{w_i(1+T(h_i))}{\sum_{i=1}^{n} w_i(1+T(h_i))}}} \right\}. \tag{14}$$

**Theorem 6.** *Let $h_i$ $(i = 1, 2, \ldots, n)$ be a collection of HFEs and $w = (w_1, w_2, \ldots, w_n)^T$ be the weight vector of $h_i$ such that $w_i \in [0, 1]$ and $\sum_{i=1}^{n} w_i = 1$, then*

$$\text{HFHPWG}_\zeta(h_1, h_2, \ldots, h_n) \geq \text{HFPWG}(h_1, h_2, \ldots, h_n).$$

**Proof.** For any $\gamma_i \in h_i$ $(i = 1, 2, \ldots, n)$, by Lemma 1, we have

$$\prod_{i=1}^{n} (1 + (\zeta - 1)(1 - \gamma_i))^{\frac{w_i(1+T(h_i))}{\sum_{i=1}^{n} w_i(1+T(h_i))}} + (\zeta - 1) \prod_{i=1}^{n} (\gamma_i)^{\frac{w_i(1+T(h_i))}{\sum_{i=1}^{n} w_i(1+T(h_i))}}$$

$$\leq \sum_{i=1}^{n} \frac{w_i(1 + T(h_i))}{\sum_{i=1}^{n} w_i(1 + T(h_i))} (1 + (\zeta - 1)(1 - \gamma_i)) + (\zeta - 1) \sum_{i=1}^{n} \frac{w_i(1 + T(h_i))}{\sum_{i=1}^{n} w_i(1 + T(h_i))} \gamma_i = \zeta.$$

Then,

$$\frac{\zeta \prod_{i=1}^{n} (\gamma_i)^{\frac{w_i(1+T(h_i))}{\sum_{i=1}^{n} w_i(1+T(h_i))}}}{\prod_{i=1}^{n} (1 + (\zeta - 1)(1 - \gamma_i))^{\frac{w_i(1+T(h_i))}{\sum_{i=1}^{n} w_i(1+T(h_i))}} + (\zeta - 1) \prod_{i=1}^{n} (\gamma_i)^{\frac{w_i(1+T(h_i))}{\sum_{i=1}^{n} w_i(1+T(h_i))}}} \geq \prod_{i=1}^{n} (\gamma_i)^{\frac{w_i(1+T(h_i))}{\sum_{i=1}^{n} w_i(1+T(h_i))}},$$

which implies that $\otimes_{Hi=1}^{n}\left(h_i^{\wedge_H \frac{w_i(1+T(h_i))}{\sum_{i=1}^{n} w_i(1+T(h_i))}}\right) \geq \otimes_{i=1}^{n}\left(h_i^{\frac{w_i(1+T(h_i))h_i}{\sum_{i=1}^{n} w_i(1+T(h_i))}}\right)$, i.e.,

$\text{HFHPWG}_\zeta(h_1, h_2, \ldots, h_n) \geq \text{HFPWG}(h_1, h_2, \ldots, h_n).$ □

Theorem 6 shows that the HFHPWG operator reflects the decision-maker's more optimistic attitude than the HFPWG operator in aggregation process. Furthermore, similar to Theorem 4, we have the properties of the HFHPWG operator as follows.

**Theorem 7.** *bLet $h_i$ $(i = 1, 2, \ldots, n)$ be a collection of HFEs and $w = (w_1, w_2, \ldots, w_n)^T$ be the weight vector of $h_i$ such that $w_i \in [0, 1]$ and $\sum_{i=1}^{n} w_i = 1$, then we have the followings:*
   *(1) Boundedness: If $h^- = \min\{\gamma_i | \gamma_i \in h_i\}$ and $h^+ = \max\{\gamma_i | \gamma_i \in h_i\}$, then*

$$h^- \leq \text{HFHPWG}_\zeta(h_1, h_2, \ldots, h_n) \leq h^+.$$

   *(2) Monotonicity: Let $h_i'$ $(i = 1, 2, \ldots, n)$ be a collection of HFEs, if $w = (w_1, w_2, \ldots, w_n)^T$ is also the weight vector of $h_i'$, and $\gamma_i \leq \gamma_i'$ for any $h_i$ and $h_i'$ $(i = 1, 2, \ldots, n)$, then*

$$\text{HFHPWG}_\zeta(h_1, h_2, \ldots, h_n) \leq \text{HFHPWG}_\zeta(h_1', h_2', \ldots, h_n').$$

**Proof.** (1) Let $g(x) = \frac{1+(\zeta-1)(1-x)}{x}$, $x \in (0, 1]$, then $g'(x) = \frac{-\zeta}{x^2} < 0$, thus $g(x)$ is a decreasing function. Since $h^- \leq \gamma_i \leq h^+$ for all $i$, then $g(h^-) \geq g(\gamma_i) \geq g(h^+)$, i.e., $\frac{1+(\zeta-1)(1-h^+)}{h^+} \leq \frac{1+(\zeta-1)(1-\gamma_i)}{\gamma_i} \leq \frac{1+(\zeta-1)(1-h^-)}{h^-}$. Since $w = (w_1, w_2, \ldots, w_n)^T$ is the weight vector of $h_i$ satisfying $w_i \in [0, 1]$ and $\sum_{i=1}^{n} w_i = 1$, then for all $i$, let $t_i = \frac{w_i(1+T(h_i))}{\sum_{i=1}^{n} w_i(1+T(h_i))}$, we have

$$\left(\frac{1+(\zeta-1)(1-h^+)}{h^+}\right)^{t_i} \leq \left(\frac{1+(\zeta-1)(1-\gamma_i)}{\gamma_i}\right)^{t_i} \leq \left(\frac{1+(\zeta-1)(1-h^-)}{h^-}\right)^{t_i}$$

$$\Leftrightarrow \prod_{i=1}^{n}\left(\frac{1+(\zeta-1)(1-h^+)}{h^+}\right)^{t_i} \leq \prod_{i=1}^{n}\left(\frac{1+(\zeta-1)(1-\gamma_i)}{\gamma_i}\right)^{t_i} \leq \prod_{i=1}^{n}\left(\frac{1+(\zeta-1)(1-h^-)}{h^-}\right)^{t_i}$$

$$\Leftrightarrow \frac{\zeta}{h^+} - (\zeta-1) \leq \prod_{i=1}^{n}\left(\frac{1+(\zeta-1)(1-\gamma_i)}{\gamma_i}\right)^{t_i} \leq \frac{\zeta}{h^-} - (\zeta-1)$$

$$\Leftrightarrow \frac{\zeta}{h^+} \leq \prod_{i=1}^{n}\left(\frac{1+(\zeta-1)(1-\gamma_i)}{\gamma_i}\right)^{t_i} + (\zeta-1) \leq \frac{\zeta}{h^-}$$

$$\Leftrightarrow \frac{h^-}{\zeta} \leq \frac{1}{\prod_{i=1}^{n}\left(\frac{1+(\zeta-1)(1-\gamma_i)}{\gamma_i}\right)^{t_i} + (\zeta-1)} \leq \frac{h^+}{\zeta}$$

$$\Leftrightarrow \frac{h^-}{\zeta} \leq \frac{\prod_{i=1}^{n}(\gamma_i)^{t_i}}{\prod_{i=1}^{n}(1+(\zeta-1)(1-\gamma_i))^{t_i} + (\zeta-1)\prod_{i=1}^{n}(\gamma_i)^{t_i}} \leq \frac{h^+}{\zeta}$$

$$\Leftrightarrow h^- \leq \frac{\zeta\prod_{i=1}^{n}(\gamma_i)^{t_i}}{\prod_{i=1}^{n}(1+(\zeta-1)(1-\gamma_i))^{t_i} + (\zeta-1)\prod_{i=1}^{n}(\gamma_i)^{t_i}} \leq h^+$$

Thus, we have $h^- \leq \text{HFHPWG}_\zeta(h_1, h_2, \ldots, h_n) \leq h^+$.

(2) Let $g(x) = \frac{1+(\zeta-1)(1-x)}{x}$, $x \in (0,1]$, then by (1), $g(x)$ is a decreasing function. Then for all $i$, $\gamma_i \leq \gamma_i'$, we have $\frac{1+(\zeta-1)(1-\gamma_i)}{\gamma_i} \geq \frac{1+(\zeta-1)(1-\gamma_i')}{\gamma_i'}$. For convenience, let $t_i = \frac{w_i(1+T(h_i))}{\sum_{i=1}^n w_i(1+T(h_i))}$, then we have

$$\left(\frac{1+(\zeta-1)(1-\gamma_i)}{\gamma_i}\right)^{t_i} \geq \left(\frac{1+(\zeta-1)(1-\gamma_i')}{\gamma_i'}\right)^{t_i}$$

$$\Leftrightarrow \prod_{i=1}^n \left(\frac{1+(\zeta-1)(1-\gamma_i)}{\gamma_i}\right)^{t_i} \geq \prod_{i=1}^n \left(\frac{1+(\zeta-1)(1-\gamma_i')}{\gamma_i'}\right)^{t_i}$$

$$\Leftrightarrow \prod_{i=1}^n \left(\frac{1+(\zeta-1)(1-\gamma_i)}{\gamma_i}\right)^{t_i} + (\zeta-1) \geq \prod_{i=1}^n \left(\frac{1+(\zeta-1)(1-\gamma_i')}{\gamma_i'}\right)^{t_i} + (\zeta-1)$$

$$\Leftrightarrow \frac{1}{\prod_{i=1}^n \left(\frac{1+(\zeta-1)(1-\gamma_i)}{\gamma_i}\right)^{t_i} + (\zeta-1)} \leq \frac{1}{\prod_{i=1}^n \left(\frac{1+(\zeta-1)(1-\gamma_i')}{\gamma_i'}\right)^{t_i} + (\zeta-1)}$$

$$\Leftrightarrow \frac{\zeta \prod_{i=1}^n (\gamma_i)^{t_i}}{\left[\begin{array}{c}\prod_{i=1}^n (1+(\zeta-1)(1-\gamma_i))^{t_i} \\ +(\zeta-1)\prod_{i=1}^n (\gamma_i)^{t_i}\end{array}\right]} \leq \frac{\zeta \prod_{i=1}^n (\gamma_i')^{t_i}}{\left[\begin{array}{c}\prod_{i=1}^n (1+(\zeta-1)(1-\gamma_i'))^{t_i} \\ +(\zeta-1)\prod_{i=1}^n (\gamma_i')^{t_i}\end{array}\right]}.$$

Thus, by Theorem 5, $\text{HFHPWG}_\zeta(h_1, h_2, \ldots, h_n) \leq \text{HFHPWG}_\zeta(h_1', h_2', \ldots, h_n')$.  □

However, the HFHPWG operator is also neither idempotent nor commutative, as illustrated by the following example.

**Example 3.** *Let $h_1$, $h_2$ and $h_3$ be three HFEs, $w$ be the weight vector of them, and $\text{Sup}(h_i, h_j)$ $(i,j = 1,2,3,$ $i \neq j)$ be the support for $h_i$ from $h_j$ given in Example 2. Then by Theorem 5, we have*

$$\text{HFHPWG}_5(h_1, h_1, h_1) = \{0.8000, 0.7095, 0.7390, 0.6455, 0.7573, 0.6644, 0.6945, 0.6000\},$$
$$\text{HFHPWG}_5(h_1, h_2, h_3) = \{0.8277, 0.6398, 0.7678, 0.5751, 0.8077, 0.6177, 0.7467, 0.5532\},$$
$$\text{HFHPWG}_5(h_2, h_3, h_1) = \{0,7881, 0.7438, 0.6611, 0.6145, 0.7445, 0.6989, 0.6152, 0.5686\}.$$

*According to Definition 3, we have $s(\text{HFHPWG}_5(h_1, h_1, h_1)) = 0.7013$, $s(\text{HFHPWG}_5(h_1, h_2, h_3)) = 0.6920$ and $s(\text{HFHPWG}_5(h_2, h_3, h_1)) = 0.6793$. Then $s(\text{HFHPWG}_5(h_1, h_1, h_1)) \neq s(h_1)$ and thus $\text{HFHPWG}_5(h_1, h_1, h_1) \neq h_1$, which implies that the HFHPWG operator is not idempotent. Furthermore, since $s(\text{HFHPWG}_5(h_1, h_2, h_3)) \neq s(\text{HFHPWG}_5(h_2, h_3, h_1))$, we have $\text{HFHPWG}_5(h_1, h_2, h_3) \neq \text{HFHPW}_5(h_2, h_3, h_1)$. Thus, the HFHPWG operator is not commutative.*

**Theorem 8.** *Let $h_i$ $(i = 1, 2, \ldots, n)$ be a collection of HFEs and $w = (w_1, w_2, \ldots, w_n)^T$ be the weight vector of $h_i$ such that $w_i \in [0,1]$ and $\sum_{i=1}^n w_i = 1$, then we have*
*(1) $\text{HFHPWA}_\zeta(h_1^c, h_2^c, \ldots, h_n^c) = (\text{HFHPWG}_\zeta(h_1, h_2, \ldots, h_n))^c$;*
*(2) $\text{HFHPWG}_\zeta(h_1^c, h_2^c, \ldots, h_n^c) = (\text{HFHPWA}_\zeta(h_1, h_2, \ldots, h_n))^c$.*

**Proof.** Since (2) is similar (1), we only prove (1).

$$\text{HFHPWA}_\zeta(h_1^c, h_2^c, \ldots, h_n^c)$$

$$= \cup_{\gamma_1 \in h_1, \gamma_2 \in h_2, \ldots, \gamma_n \in h_n} \left\{ \frac{\prod_{i=1}^n \left(1 + (\zeta - 1)(1 - \gamma_i)\right)^{\frac{w_i(1+T(h_i))}{\sum_{i=1}^n w_i(1+T(h_i))}} - \prod_{i=1}^n (\gamma_i)^{\frac{w_i(1+T(h_i))}{\sum_{i=1}^n w_i(1+T(h_i))}}}{\prod_{i=1}^n \left(1 + (\zeta - 1)(1 - \gamma_i)\right)^{\frac{w_i(1+T(h_i))}{\sum_{i=1}^n w_i(1+T(h_i))}} + (\zeta - 1)\prod_{i=1}^n (\gamma_i)^{\frac{w_i(1+T(h_i))}{\sum_{i=1}^n w_i(1+T(h_i))}}} \right\}$$

$$= \cup_{\gamma_1 \in h_1, \gamma_2 \in h_2, \ldots, \gamma_n \in h_n} \left\{ 1 - \frac{\zeta \prod_{i=1}^n (\gamma_i)^{\frac{w_i(1+T(h_i))}{\sum_{i=1}^n w_i(1+T(h_i))}}}{\left[ \begin{array}{c} \prod_{i=1}^n \left(1 + (\zeta - 1)(1 - \gamma_i)\right)^{\frac{w_i(1+T(h_i))}{\sum_{i=1}^n w_i(1+T(h_i))}} \\ + (\zeta - 1)\prod_{i=1}^n (\gamma_i)^{\frac{w_i(1+T(h_i))}{\sum_{i=1}^n w_i(1+T(h_i))}} \end{array} \right]} \right\}$$

$$= (\text{HFHPWG}_\zeta(h_1, h_2, \ldots, h_n))^c.$$

□

*3.2. Generalized Hesitant Fuzzy Hamacher Power-Weighted Average/Geometric Operators*

**Definition 7.** *Let $h_i$ ($i = 1, 2, \ldots, n$) be a collection of HFEs, $w = (w_1, w_2, \ldots, w_n)^T$ be the weight vector of $h_i$ ($i = 1, 2, \ldots, n$) such that $w_i \in [0, 1]$ and $\sum_{i=1}^n w_i = 1$. For a parameter $\lambda > 0$, a generalized hesitant fuzzy Hamacher power-weighted average (GHFHPWA) operator is a function $H^n \to H$ such that*

$$\text{GHFHPWA}_\zeta(h_1, h_2, \ldots, h_n) = \left( \oplus_{H_{i=1}}^n \left( \frac{w_i(1 + T(h_i)) \cdot_H h_i^{\wedge_H \lambda}}{\sum_{i=1}^n w_i(1 + T(h_i))} \right) \right)^{\wedge_H \frac{1}{\lambda}}, \tag{15}$$

*where parameter $\zeta > 0$, $T(h_i) = \sum_{j=1, j \neq i}^n w_j \text{Sup}(h_i, h_j)$ and $\text{Sup}(h_i, h_j)$ is the support for $h_i$ from $h_j$.*

**Theorem 9.** *Let $h_i$ ($i = 1, 2, \ldots, n$) be a collection of HFEs and $w = (w_1, w_2, \ldots, w_n)^T$ be the weight vector of $h_i$ ($i = 1, 2, \ldots, n$) such that $w_i \in [0, 1]$ and $\sum_{i=1}^n w_i = 1$, then the aggregated value by GHFHPWA operator is also a HFE, and*

$$\text{GHFHPWA}_\zeta(h_1, h_2, \ldots, h_n)$$

$$= \cup_{\gamma_1 \in h_1, \gamma_2 \in h_2, \ldots, \gamma_n \in h_n} \left\{ \frac{\zeta \left( \prod_{i=1}^n a_i^{\frac{w_i(1+T(h_i))}{\sum_{i=1}^n w_i(1+T(h_i))}} - \prod_{i=1}^n b_i^{\frac{w_i(1+T(h_i))}{\sum_{i=1}^n w_i(1+T(h_i))}} \right)^{\frac{1}{\lambda}}}{\left[ \begin{array}{c} \left( \prod_{i=1}^n a_i^{\frac{w_i(1+T(h_i))}{\sum_{i=1}^n w_i(1+T(h_i))}} + (\zeta^2 - 1)\prod_{i=1}^n b_i^{\frac{w_i(1+T(h_i))}{\sum_{i=1}^n w_i(1+T(h_i))}} \right)^{\frac{1}{\lambda}} \\ + (\zeta - 1)\left( \prod_{i=1}^n a_i^{\frac{w_i(1+T(h_i))}{\sum_{i=1}^n w_i(1+T(h_i))}} - \prod_{i=1}^n b_i^{\frac{w_i(1+T(h_i))}{\sum_{i=1}^n w_i(1+T(h_i))}} \right)^{\frac{1}{\lambda}} \end{array} \right]} \right\}, \tag{16}$$

*where $a_i = (1 + (\zeta - 1)(1 - \gamma_i))^\lambda + (\zeta^2 - 1)\gamma_i^\lambda$ and $b_i = (1 + (\zeta - 1)(1 - \gamma_i))^\lambda - \gamma_i^\lambda$.*

**Proof.** We first use the mathematical induction on $n$ to prove

$$\oplus_{H_{i=1}^{n}} \left( \frac{w_i(1 + T(h_i)) \cdot_H h_i^{\wedge_H \lambda}}{\sum_{i=1}^{n} w_i(1 + T(h_i))} \right)$$

$$= \cup_{\gamma_1 \in h_1, \gamma_2 \in h_2, \dots, \gamma_n \in h_n} \left\{ \frac{\prod_{i=1}^{n} a_i^{\frac{w_i(1+T(h_i))}{\sum_{i=1}^{n} w_i(1+T(h_i))}} - \prod_{i=1}^{n} b_i^{\frac{w_i(1+T(h_i))}{\sum_{i=1}^{n} w_i(1+T(h_i))}}}{\prod_{i=1}^{n} a_i^{\frac{w_i(1+T(h_i))}{\sum_{i=1}^{n} w_i(1+T(h_i))}} + (\zeta - 1) \prod_{i=1}^{n} b_i^{\frac{w_i(1+T(h_i))}{\sum_{i=1}^{n} w_i(1+T(h_i))}}} \right\}. \tag{17}$$

(1) When $n = 1$, since $\frac{w_i(1+T(h_i))}{\sum_{i=1}^{n} w_i(1+T(h_i))} = 1$, we have

$$\oplus_{H_{i=1}^{n}} \left( \frac{w_i(1 + T(h_i)) \cdot_H h_i^{\wedge_H \lambda}}{\sum_{i=1}^{n} w_i(1 + T(h_i))} \right) = h_1^{\wedge_H \lambda}$$

$$= \cup_{\gamma_1 \in h_1} \left\{ \frac{\zeta \gamma_1^\lambda}{(1 + (\zeta - 1)(1 - \gamma_1))^\lambda + (\zeta - 1)\gamma_1^\lambda} \right\}$$

$$= \cup_{\gamma_1 \in h_1} \left\{ \frac{a_1 - b_1}{a_1 + (\zeta - 1)b_1} \right\}.$$

Thus, Equation (17) holds for $n = 1$.
(2) Suppose that Equation (17) holds for $n = k$, that is

$$\oplus_{H_{i=1}^{k}} \left( \frac{w_i(1 + T(h_i)) \cdot_H h_i^{\wedge_H \lambda}}{\sum_{i=1}^{k} w_i(1 + T(h_i))} \right)$$

$$= \cup_{\gamma_1 \in h_1, \gamma_2 \in h_2, \dots, \gamma_k \in h_k} \left\{ \frac{\prod_{i=1}^{k} a_i^{\frac{w_i(1+T(h_i))}{\sum_{i=1}^{k} w_i(1+T(h_i))}} - \prod_{i=1}^{k} b_i^{\frac{w_i(1+T(h_i))}{\sum_{i=1}^{k} w_i(1+T(h_i))}}}{\prod_{i=1}^{k} a_i^{\frac{w_i(1+T(h_i))}{\sum_{i=1}^{k} w_i(1+T(h_i))}} + (\zeta - 1) \prod_{i=1}^{k} b_i^{\frac{w_i(1+T(h_i))}{\sum_{i=1}^{k} w_i(1+T(h_i))}}} \right\},$$

then, when $n = k + 1$, by the operational laws in Definition 4, we have

$$\oplus_{H_{i=1}^{k+1}} \left( \frac{w_i(1 + T(h_i)) \cdot_H h_i^{\wedge_H \lambda}}{\sum_{i=1}^{k+1} w_i(1 + T(h_i))} \right)$$

$$= \oplus_{H_{i=1}^{k}} \left( \frac{w_i(1 + T(h_i)) \cdot_H h_i^{\wedge_H \lambda}}{\sum_{i=1}^{k+1} w_i(1 + T(h_i))} \right) \oplus_H \left( \frac{w_{k+1}(1 + T(h_{k+1})) \cdot_H h_{k+1}^{\wedge_H \lambda}}{\sum_{i=1}^{k+1} w_i(1 + T(h_i))} \right)$$

$$= \cup_{\gamma_1 \in h_1, \gamma_2 \in h_2, \dots, \gamma_k \in h_k} \left\{ \frac{\prod_{i=1}^{k} a_i^{\frac{w_i(1+T(h_i))}{\sum_{i=1}^{k+1} w_i(1+T(h_i))}} - \prod_{i=1}^{k} b_i^{\frac{w_i(1+T(h_i))}{\sum_{i=1}^{k+1} w_i(1+T(h_i))}}}{\prod_{i=1}^{k} a_i^{\frac{w_i(1+T(h_i))}{\sum_{i=1}^{k+1} w_i(1+T(h_i))}} + (\zeta - 1) \prod_{i=1}^{k} b_i^{\frac{w_i(1+T(h_i))}{\sum_{i=1}^{k+1} w_i(1+T(h_i))}}} \right\}$$

$$\oplus_H \cup_{\gamma_{k+1} \in h_{k+1}} \left\{ \frac{a_{k+1}^{\frac{w_{k+1}(1+T(h_{k+1}))}{\sum_{i=1}^{k+1} w_i(1+T(h_i))}} - b_{k+1}^{\frac{w_{k+1}(1+T(h_{k+1}))}{\sum_{i=1}^{k+1} w_i(1+T(h_i))}}}{a_{k+1}^{\frac{w_{k+1}(1+T(h_{k+1}))}{\sum_{i=1}^{k+1} w_i(1+T(h_i))}} + (\zeta - 1) b_{k+1}^{\frac{w_{k+1}(1+T(h_{k+1}))}{\sum_{i=1}^{k+1} w_i(1+T(h_i))}}} \right\}$$

$$= \cup_{\gamma_1 \in h_1, \gamma_2 \in h_2, \dots, \gamma_k \in h_k, \gamma_{k+1} \in h_{k+1}} \left\{ \frac{\prod_{i=1}^{k+1} a_i^{\frac{w_i(1+T(h_i))}{\sum_{i=1}^{k+1} w_i(1+T(h_i))}} - \prod_{i=1}^{k+1} b_i^{\frac{w_i(1+T(h_i))}{\sum_{i=1}^{k+1} w_i(1+T(h_i))}}}{\prod_{i=1}^{k+1} a_i^{\frac{w_i(1+T(h_i))}{\sum_{i=1}^{k+1} w_i(1+T(h_i))}} + (\zeta - 1) \prod_{i=1}^{k+1} b_i^{\frac{w_i(1+T(h_i))}{\sum_{i=1}^{k+1} w_i(1+T(h_i))}}} \right\},$$

i.e., Equation (17) holds for $n = k + 1$. Thus, Equation (17) holds for all $n$.

Hence, by the operational laws in Definition 4, we have

$$\text{GHFHPWA}_\zeta(h_1, h_2, \ldots, h_n) = \left( \oplus_{H_{i=1}}^{n} \left( \frac{w_i(1 + T(h_i)) \cdot_H h_i^{\wedge_H \lambda}}{\sum_{i=1}^{n} w_i(1 + T(h_i))} \right) \right)^{\wedge_H \frac{1}{\lambda}}$$

$$= \cup_{\gamma_1 \in h_1, \gamma_2 \in h_2, \ldots, \gamma_n \in h_n} \left\{ \frac{\zeta \left( \frac{\prod_{i=1}^{n} a_i^{\frac{w_i(1+T(h_i))}{\sum_{i=1}^{n} w_i(1+T(h_i))}} - \prod_{i=1}^{n} b_i^{\frac{w_i(1+T(h_i))}{\sum_{i=1}^{n} w_i(1+T(h_i))}}}{\prod_{i=1}^{n} a_i^{\frac{w_i(1+T(h_i))}{\sum_{i=1}^{n} w_i(1+T(h_i))}} + (\zeta-1) \prod_{i=1}^{n} b_i^{\frac{w_i(1+T(h_i))}{\sum_{i=1}^{n} w_i(1+T(h_i))}}} \right)^{\frac{1}{\lambda}}}{\left[ \left( 1 + (\zeta-1)(1 - \frac{\prod_{i=1}^{n} a_i^{\frac{w_i(1+T(h_i))}{\sum_{i=1}^{n} w_i(1+T(h_i))}} - \prod_{i=1}^{n} b_i^{\frac{w_i(1+T(h_i))}{\sum_{i=1}^{n} w_i(1+T(h_i))}}}{\prod_{i=1}^{n} a_i^{\frac{w_i(1+T(h_i))}{\sum_{i=1}^{n} w_i(1+T(h_i))}} + (\zeta-1) \prod_{i=1}^{n} b_i^{\frac{w_i(1+T(h_i))}{\sum_{i=1}^{n} w_i(1+T(h_i))}}}) \right)^{\frac{1}{\lambda}} + (\zeta-1) \left( \frac{\prod_{i=1}^{n} a_i^{\frac{w_i(1+T(h_i))}{\sum_{i=1}^{n} w_i(1+T(h_i))}} - \prod_{i=1}^{n} b_i^{\frac{w_i(1+T(h_i))}{\sum_{i=1}^{n} w_i(1+T(h_i))}}}{\prod_{i=1}^{n} a_i^{\frac{w_i(1+T(h_i))}{\sum_{i=1}^{n} w_i(1+T(h_i))}} + (\zeta-1) \prod_{i=1}^{n} b_i^{\frac{w_i(1+T(h_i))}{\sum_{i=1}^{n} w_i(1+T(h_i))}}} \right)^{\frac{1}{\lambda}} \right]} \right\}$$

$$= \cup_{\gamma_1 \in h_1, \gamma_2 \in h_2, \ldots, \gamma_n \in h_n} \left\{ \frac{\zeta \left( \prod_{i=1}^{n} a_i^{\frac{w_i(1+T(h_i))}{\sum_{i=1}^{n} w_i(1+T(h_i))}} - \prod_{i=1}^{n} b_i^{\frac{w_i(1+T(h_i))}{\sum_{i=1}^{n} w_i(1+T(h_i))}} \right)^{\frac{1}{\lambda}}}{\left[ \left( \prod_{i=1}^{n} a_i^{\frac{w_i(1+T(h_i))}{\sum_{i=1}^{n} w_i(1+T(h_i))}} + (\zeta^2 - 1) \prod_{i=1}^{n} b_i^{\frac{w_i(1+T(h_i))}{\sum_{i=1}^{n} w_i(1+T(h_i))}} \right)^{\frac{1}{\lambda}} + (\zeta-1) \left( \prod_{i=1}^{n} a_i^{\frac{w_i(1+T(h_i))}{\sum_{i=1}^{n} w_i(1+T(h_i))}} - \prod_{i=1}^{n} b_i^{\frac{w_i(1+T(h_i))}{\sum_{i=1}^{n} w_i(1+T(h_i))}} \right)^{\frac{1}{\lambda}} \right]} \right\},$$

which completes the proof of the theorem. $\square$

**Remark 3.** *(1) If* $\text{Sup}(h_i, h_j) = k$, *for all* $i \neq j$, *then*

$$\text{GHFHPWA}_\zeta(h_1, h_2, \ldots, h_n) = \left( \oplus_{H_{i=1}}^{n} \left( w_i \cdot_H h_i^{\wedge_H \lambda} \right) \right)^{\wedge_H \frac{1}{\lambda}} \tag{18}$$

*and thus, the GHFHPWA operator reduces to the generalized hesitant fuzzy Hamacher weighted average (GHFHWA) operator [17].*

*(2) If* $\zeta = 1$, *then the GHFHPWA operator reduces to the generalized hesitant fuzzy power-weighted average (GHFPWA) operator [15]:*

$$\text{GHFHPWA}_1(h_1, h_2, \ldots, h_n) = \cup_{\gamma_1 \in h_1, \gamma_2 \in h_2, \ldots, \gamma_n \in h_n} \left\{ \left( 1 - \prod_{i=1}^{n} \left( 1 - \gamma_i^\lambda \right)^{\frac{w_i(1+T(h_i))}{\sum_{i=1}^{n} w_i(1+T(h_i))}} \right)^{\frac{1}{\lambda}} \right\} \tag{19}$$

*and if $\zeta = 2$, then the GHFHPWA operator reduces to the generalized hesitant fuzzy Einstein power-weighted average (GHFEPWA) operator [18]:*

$$
\begin{aligned}
&\text{GHFHPWA}_2(h_1, h_2, \ldots, h_n) \\
&= \cup_{\gamma_1 \in h_1, \gamma_2 \in h_2, \ldots, \gamma_n \in h_n} \left\{ \frac{2\left( \prod_{i=1}^n a_i^{\frac{w_i(1+T(h_i))}{\sum_{i=1}^n w_i(1+T(h_i))}} - \prod_{i=1}^n b_i^{\frac{w_i(1+T(h_i))}{\sum_{i=1}^n w_i(1+T(h_i))}} \right)^{\frac{1}{\lambda}}}{\left[ \left( \prod_{i=1}^n a_i^{\frac{w_i(1+T(h_i))}{\sum_{i=1}^n w_i(1+T(h_i))}} + 3\prod_{i=1}^n b_i^{\frac{w_i(1+T(h_i))}{\sum_{i=1}^n w_i(1+T(h_i))}} \right)^{\frac{1}{\lambda}} + \left( \prod_{i=1}^n a_i^{\frac{w_i(1+T(h_i))}{\sum_{i=1}^n w_i(1+T(h_i))}} - \prod_{i=1}^n b_i^{\frac{w_i(1+T(h_i))}{\sum_{i=1}^n w_i(1+T(h_i))}} \right)^{\frac{1}{\lambda}} \right]} \right\}, \quad (20)
\end{aligned}
$$

*where $a_i = (2 - \gamma_i)^\lambda + 3\gamma_i^\lambda$, $b_i = (2 - \gamma_i)^\lambda - \gamma_i^\lambda$.*

(3) *If $\lambda = 1$, then $a_i = \zeta(1 + (\zeta - 1)\gamma_i)$ and $b_i = \zeta(1 - \gamma_i)$, and thus the GHFHPWA operator reduces to the HFHPWA operator.*

*In particular, if $w = (\frac{1}{n}, \frac{1}{n}, \ldots, \frac{1}{n})^T$, then the GHFHPWA operator reduces to the generalized hesitant fuzzy Hamacher power average (GHFHPA) operator:*

$$
\begin{aligned}
&\text{GHFHPA}_\zeta(h_1, h_2, \ldots, h_n) = \left( \oplus_{H_{i=1}}^n \left( \frac{(1 + T'(h_i)) \cdot_H h_i^{\wedge_H \lambda}}{\sum_{i=1}^n (1 + T'(h_i))} \right) \right)^{\wedge_H \frac{1}{\lambda}} \\
&= \cup_{\gamma_1 \in h_1, \gamma_2 \in h_2, \ldots, \gamma_n \in h_n} \left\{ \frac{\zeta\left( \prod_{i=1}^n a_i^{\frac{(1+T'(h_i))}{\sum_{i=1}^n (1+T'(h_i))}} - \prod_{i=1}^n b_i^{\frac{(1+T'(h_i))}{\sum_{i=1}^n (1+T'(h_i))}} \right)^{\frac{1}{\lambda}}}{\left[ \left( \prod_{i=1}^n a_i^{\frac{(1+T'(h_i))}{\sum_{i=1}^n (1+T'(h_i))}} + (\zeta^2 - 1)\prod_{i=1}^n b_i^{\frac{(1+T'(h_i))}{\sum_{i=1}^n (1+T'(h_i))}} \right)^{\frac{1}{\lambda}} + (\zeta - 1)\left( \prod_{i=1}^n a_i^{\frac{w_i(1+T(h_i))}{\sum_{i=1}^n w_i(1+T(h_i))}} - \prod_{i=1}^n b_i^{\frac{w_i(1+T(h_i))}{\sum_{i=1}^n w_i(1+T(h_i))}} \right)^{\frac{1}{\lambda}} \right]} \right\}, \quad (21)
\end{aligned}
$$

*where $a_i = (1 + (\zeta - 1)(1 - \gamma_i))^\lambda + (\zeta^2 - 1)\gamma_i^\lambda$, $b_i = (1 + (\zeta - 1)(1 - \gamma_i))^\lambda - \gamma_i^\lambda$ and $T'(h_i) = \frac{1}{n}\sum_{j=1, j\neq i}^n \text{Sup}(h_i, h_j)$.*

**Definition 8.** *Let $h_i$ ($i = 1, 2, \ldots, n$) be a collection of HFEs and $w = (w_1, w_2, \ldots, w_n)^T$ be the weight vector of $h_i$ ($i = 1, 2, \ldots, n$) such that $w_i \in [0, 1]$ and $\sum_{i=1}^n w_i = 1$. For $\lambda > 0$, a generalized hesitant fuzzy Hamacher power-weighted geometric (GHFHPWG) operator is a function $H^n \to H$ such that*

$$
\text{GHFHPWG}_\zeta(h_1, h_2, \ldots, h_n) = \frac{1}{\lambda} \cdot_H \left( \otimes_{H_{i=1}}^n \left( (\lambda \cdot_H h_i)^{\wedge_H \frac{w_i(1+T(h_i))}{\sum_{i=1}^n w_i(1+T(h_i))}} \right) \right), \quad (22)
$$

*where $\zeta > 0$, $T(h_i) = \sum_{j=1, j\neq i}^n w_j\text{Sup}(h_i, h_j)$ and $\text{Sup}(h_i, h_j)$ is the support for $h_i$ from $h_j$.*

**Theorem 10.** *Let $h_i$ $(i = 1, 2, \ldots, n)$ be a collection of HFEs and $w = (w_1, w_2, \ldots, w_n)^T$ be the weight vector of $h_i$ $(i = 1, 2, \ldots, n)$ such that $w_i \in [0, 1]$ and $\sum_{i=1}^n w_i = 1$, then the aggregated value by GHFHPWG operator is also a HFE, and*

$$\text{GHFHPWG}_\zeta(h_1, h_2, \ldots, h_n)$$

$$= \cup_{\gamma_1 \in h_1, \gamma_2 \in h_2, \ldots, \gamma_n \in h_n} \left\{ \frac{\left[ \begin{array}{c} \left( \prod_{i=1}^n c_i^{\frac{w_i(1+T(h_i))}{\sum_{i=1}^n w_i(1+T(h_i))}} + (\zeta^2 - 1) \prod_{i=1}^n d_i^{\frac{w_i(1+T(h_i))}{\sum_{i=1}^n w_i(1+T(h_i))}} \right)^{\frac{1}{\lambda}} \\ - \left( \prod_{i=1}^n c_i^{\frac{w_i(1+T(h_i))}{\sum_{i=1}^n w_i(1+T(h_i))}} - \prod_{i=1}^n d_i^{\frac{w_i(1+T(h_i))}{\sum_{i=1}^n w_i(1+T(h_i))}} \right)^{\frac{1}{\lambda}} \end{array} \right]}{\left[ \begin{array}{c} \left( \prod_{i=1}^n c_i^{\frac{w_i(1+T(h_i))}{\sum_{i=1}^n w_i(1+T(h_i))}} + (\zeta^2 - 1) \prod_{i=1}^n d_i^{\frac{w_i(1+T(h_i))}{\sum_{i=1}^n w_i(1+T(h_i))}} \right)^{\frac{1}{\lambda}} \\ + (\zeta - 1) \left( \prod_{i=1}^n c_i^{\frac{w_i(1+T(h_i))}{\sum_{i=1}^n w_i(1+T(h_i))}} - \prod_{i=1}^n d_i^{\frac{w_i(1+T(h_i))}{\sum_{i=1}^n w_i(1+T(h_i))}} \right)^{\frac{1}{\lambda}} \end{array} \right]} \right\}, \quad (23)$$

*where $c_i = (1 + (\zeta - 1)\gamma_i)^\lambda + (\zeta^2 - 1)(1 - \gamma_i)^\lambda$ and $d_i = (1 + (\zeta - 1)\gamma_i)^\lambda - (1 - \gamma_i)^\lambda$.*

**Proof.** Similar to the proof of Theorem 9, Equation (23) can be proved by mathematical induction on $n$. □

In particular, if $w = (\frac{1}{n}, \frac{1}{n}, \ldots, \frac{1}{n})^T$, then the GHFHPWG operator reduces to the generalized hesitant fuzzy Hamacher power geometric (GHFHPG) operator:

$$\text{GHFHPA}_\zeta(h_1, h_2, \ldots, h_n) = \frac{1}{\lambda} \cdot_H \left( \otimes_{H i=1}^n \left( (\lambda \cdot_H h_i)^{\wedge_H \frac{(1+T'(h_i))}{\sum_{i=1}^n (1+T'(h_i))}} \right) \right)$$

$$= \cup_{\gamma_1 \in h_1, \gamma_2 \in h_2, \ldots, \gamma_n \in h_n} \left\{ \frac{\left[ \begin{array}{c} \left( \prod_{i=1}^n c_i^{\frac{(1+T'(h_i))}{\sum_{i=1}^n (1+T'(h_i))}} + (\zeta^2 - 1) \prod_{i=1}^n d_i^{\frac{(1+T'(h_i))}{\sum_{i=1}^n (1+T'(h_i))}} \right)^{\frac{1}{\lambda}} \\ - \left( \prod_{i=1}^n c_i^{\frac{(1+T'(h_i))}{\sum_{i=1}^n (1+T'(h_i))}} - \prod_{i=1}^n d_i^{\frac{(1+T'(h_i))}{\sum_{i=1}^n (1+T'(h_i))}} \right)^{\frac{1}{\lambda}} \end{array} \right]}{\left[ \begin{array}{c} \left( \prod_{i=1}^n c_i^{\frac{(1+T'(h_i))}{\sum_{i=1}^n (1+T'(h_i))}} + (\zeta^2 - 1) \prod_{i=1}^n d_i^{\frac{(1+T'(h_i))}{\sum_{i=1}^n (1+T'(h_i))}} \right)^{\frac{1}{\lambda}} \\ + (\zeta - 1) \left( \prod_{i=1}^n c_i^{\frac{(1+T'(h_i))}{\sum_{i=1}^n (1+T'(h_i))}} - \prod_{i=1}^n d_i^{\frac{(1+T'(h_i))}{\sum_{i=1}^n (1+T'(h_i))}} \right)^{\frac{1}{\lambda}} \end{array} \right]} \right\}, \quad (24)$$

*where* $c_i = (1 + (\zeta - 1)\gamma_i)^\lambda + (\zeta^2 - 1)(1 - \gamma_i)^\lambda$, $d_i = (1 + (\zeta - 1)\gamma_i)^\lambda - (1 - \gamma_i)^\lambda$ *and* $T'(h_i) = \frac{1}{n} \sum_{j=1, j \neq i}^n \text{Sup}(h_i, h_j)$.

**Remark 4.** *(1) If $\zeta = 1$, then the GHFHPWG operator reduces to the generalized hesitant fuzzy power-weighted geometric (GHFPWG) operator [15]:*

$$\text{GHFHPWG}_1(h_1, h_2, \ldots, h_n) = \cup_{\gamma_1 \in h_1, \gamma_2 \in h_2, \ldots, \gamma_n \in h_n} \left\{ \left( \prod_{i=1}^n \left( \gamma_i^\lambda \right)^{\frac{w_i(1+T(h_i))}{\sum_{i=1}^n w_i(1+T(h_i))}} \right)^{\frac{1}{\lambda}} \right\}. \quad (25)$$

*and if $\zeta = 2$, then the GHFHPWG operator reduces to the generalized hesitant fuzzy Einstein power-weighted geometric (GHFEPWG) operator [18]:*

$$\text{GHFHPWG}_2(h_1, h_2, \ldots, h_n)$$

$$= \cup_{\gamma_1 \in h_1, \gamma_2 \in h_2, \ldots, \gamma_n \in h_n} \left\{ \frac{\left[ \left( \prod_{i=1}^n c_i^{\frac{w_i(1+T(h_i))}{\sum_{i=1}^n w_i(1+T(h_i))}} + 3 \prod_{i=1}^n d_i^{\frac{w_i(1+T(h_i))}{\sum_{i=1}^n w_i(1+T(h_i))}} \right)^{\frac{1}{\lambda}} - \left( \prod_{i=1}^n c_i^{\frac{w_i(1+T(h_i))}{\sum_{i=1}^n w_i(1+T(h_i))}} - \prod_{i=1}^n d_i^{\frac{w_i(1+T(h_i))}{\sum_{i=1}^n w_i(1+T(h_i))}} \right)^{\frac{1}{\lambda}} \right]}{\left[ \left( \prod_{i=1}^n c_i^{\frac{w_i(1+T(h_i))}{\sum_{i=1}^n w_i(1+T(h_i))}} + 3 \prod_{i=1}^n d_i^{\frac{w_i(1+T(h_i))}{\sum_{i=1}^n w_i(1+T(h_i))}} \right)^{\frac{1}{\lambda}} + (\zeta - 1) \left( \prod_{i=1}^n c_i^{\frac{w_i(1+T(h_i))}{\sum_{i=1}^n w_i(1+T(h_i))}} - \prod_{i=1}^n d_i^{\frac{w_i(1+T(h_i))}{\sum_{i=1}^n w_i(1+T(h_i))}} \right)^{\frac{1}{\lambda}} \right]} \right\}, \quad (26)$$

*where $c_i = (1 + \gamma_i)^\lambda + 3(1 - \gamma_i)^\lambda$, $d_i = (1 + \gamma_i)^\lambda - (1 - \gamma_i)^\lambda$. (2) If $\text{Sup}(h_i, h_j) = k$, for all $i \neq j$, then*

$$\text{GHFHPWG}_\zeta(h_1, h_2, \ldots, h_n) = \frac{1}{\lambda} \cdot_H \left( \otimes_H \big|_{i=1}^n (\lambda \cdot_H h_i)^{\wedge_H w_i} \right) \quad (27)$$

*and thus, the GHFHPWG operator reduces to the generalized hesitant fuzzy Hamacher weighted geometric (GHFHWG) operator [17].*

*(3) If $\lambda = 1$, then $c_i = \zeta(1 + (\zeta - 1)(1 - \gamma_i))$ and $d_i = \zeta \gamma_i$, and so the GHFHPWG operator reduces to the HFHPWG operator.*

**Theorem 11.** *Let $h_i$ $(i = 1, 2, \ldots, n)$ be a collection of HFEs and $w = (w_1, w_2, \ldots, w_n)^T$ be the weight vector of $h_i$ such that $w_i \in [0, 1]$ and $\sum_{i=1}^n w_i = 1$, then we have*
*(1) $\text{GHFHPWA}_\zeta(h_1^c, h_2^c, \ldots, h_n^c) = (\text{GHFHPWG}_\zeta(h_1, h_2, \ldots, h_n))^c$;*
*(2) $\text{GHFHPWG}_\zeta(h_1^c, h_2^c, \ldots, h_n^c) = (\text{GHFHPWA}_\zeta(h_1, h_2, \ldots, h_n))^c$.*

**Proof.** Similar to the proof of Theorem 8. □

*3.3. Hesitant Fuzzy Hamacher Power-Ordered Weighted Average/Geometric Operators*

Motivated by the idea of the POWA operator [24], POWG operator [25] and Hamacher operations, we define the hesitant fuzzy Hamacher power-ordered weighted average (HFHPOWA) operator and hesitant fuzzy Hamacher power-ordered weighted geometric (HFHPOWG) operator as follows.

**Definition 9.** *Let $h_i$ $(i = 1, 2, \ldots, n)$ be a collection of HFEs. A hesitant fuzzy Hamacher power-ordered weighted average (HFHPOWA) operator is a function $H^n \to H$ such that*

$$\text{HFHPOWA}_\zeta(h_1, h_2, \ldots, h_n) = \oplus_H \big|_{i=1}^n \left( u_i \cdot_H h_{\sigma(i)} \right), \quad (28)$$

*where parameter $\zeta > 0$, $h_{\sigma(i)}$ is the ith largest HFE of $h_j$ $(j = 1, 2, \ldots, n)$, and $u_i$ $(i = 1, 2, \ldots, n)$ is a collection of weights such that*

$$u_i = g\left( \frac{R_i}{TV} \right) - g\left( \frac{R_{i-1}}{TV} \right), \ R_i = \sum_{j=1}^i V_{\sigma(j)}, \ TV = \sum_{i=1}^n V_{\sigma(i)},$$

$$V_{\sigma(i)} = 1 + T(h_{\sigma(i)}), \ T(h_{\sigma(i)}) = \sum_{j=1, j \neq i}^n \text{Sup}(h_{\sigma(i)}, h_{\sigma(j)}), \quad (29)$$

where $T(h_{\sigma(i)})$ denotes the support of *i*th largest HFE by all of the other HFEs, $\mathrm{Sup}(h_{\sigma(i)}, h_{\sigma(j)})$ indicates the support of the *i*th largest HFE for the *j*th largest HFE, and $g : [0,1] \to [0,1]$ is a basic unit-interval monotone (BUM) function with the following properties: (1) $g(0) = 0$; (2) $g(1) = 1$; and (3) $g(x) \geq g(y)$ if $x > y$.

**Theorem 12.** *Let $h_i$ ($i = 1, 2, \ldots, n$) be a collection of HFEs, then the aggregated value by HFHPOWA operator is also an HFE, and*

$$\mathrm{HFHPOWA}_\zeta(h_1, h_2, \ldots, h_n)$$

$$= \cup_{\gamma_{\sigma(1)} \in h_{\sigma(1)}, \gamma_{\sigma(2)} \in h_{\sigma(2)}, \ldots, \gamma_{\sigma(n)} \in h_{\sigma(n)}} \left\{ \frac{\prod_{i=1}^n \left(1 + (\zeta - 1)\gamma_{\sigma(i)}\right)^{u_i} - \prod_{i=1}^n \left(1 - \gamma_{\sigma(i)}\right)^{u_i}}{\prod_{i=1}^n \left(1 + (\zeta - 1)\gamma_{\sigma(i)}\right)^{u_i} + (\zeta - 1) \prod_{i=1}^n \left(1 - \gamma_{\sigma(i)}\right)^{u_i}} \right\}, \quad (30)$$

*where $u_i$ ($i = 1, 2, \ldots, n$) is a collection of weights satisfying the condition (29).*

**Remark 5.** *(1) If $\mathrm{Sup}(h_i, h_j) = k$ for all $i \neq j$ and $g(x) = x$, then $u_i = \frac{1}{n}$, $i = 1, 2, \ldots, n$, and so*

$$\mathrm{HFHPOWA}_\zeta(h_1, h_2, \ldots, h_n) = \oplus_{H_{i=1}^n} \left( \frac{1}{n} \cdot_H h_i \right)$$

*which indicates that the HFHPOWA operator reduces to the hesitant fuzzy Hamacher average (HFHA) operator [17].*

*(2) If $\zeta = 1$, then the HFHPOWA operator reduces to the hesitant fuzzy power-ordered weighted average (HFPOWA) operator [15]:*

$$\mathrm{HFHPOWA}_1(h_1, h_2, \ldots, h_n) = \oplus_{i=1}^n \left( u_i h_{\sigma(i)} \right)$$

$$= \cup_{\gamma_{\sigma(1)} \in h_{\sigma(1)}, \gamma_{\sigma(2)} \in h_{\sigma(2)}, \ldots, \gamma_{\sigma(n)} \in h_{\sigma(n)}} \left\{ 1 - \prod_{i=1}^n \left(1 - \gamma_{\sigma(i)}\right)^{u_i} \right\}, \quad (31)$$

*where $u_i$ ($i = 1, 2, \ldots, n$) is a collection of weights satisfying the condition (29). If $\zeta = 2$, then the HFHPOWA operator (30) reduces to the hesitant fuzzy Einstein power-ordered weighted average (HFEPOWA) operator [18]:*

$$\mathrm{HFHPOWA}_2(h_1, h_2, \ldots, h_n)$$

$$= \cup_{\gamma_{\sigma(1)} \in h_{\sigma(1)}, \gamma_{\sigma(2)} \in h_{\sigma(2)}, \ldots, \gamma_{\sigma(n)} \in h_{\sigma(n)}} \left\{ \frac{\prod_{i=1}^n \left(1 + \gamma_{\sigma(i)}\right)^{u_i} - \prod_{i=1}^n \left(1 - \gamma_{\sigma(i)}\right)^{u_i}}{\prod_{i=1}^n \left(1 + \gamma_{\sigma(i)}\right)^{u_i} + \prod_{i=1}^n \left(1 - \gamma_{\sigma(i)}\right)^{u_i}} \right\}, \quad (32)$$

*where $u_i$ ($i = 1, 2, \ldots, n$) is a collection of weights satisfying the condition (29).*

Similar to Theorems 3 and 4, we have the properties of HFHPOWA operator as follows.

**Theorem 13.** *If $h_i$ ($i = 1, 2, \ldots, n$) is a collection of HFEs and $u_i$ ($i = 1, 2, \ldots, n$) is the collection of the weights which satisfies the condition (29), then*

$$\mathrm{HFHPOWA}_\zeta(h_1, h_2, \ldots, h_n) \leq \mathrm{HFPOWA}(h_1, h_2, \ldots, h_n).$$

**Theorem 14.** *If $h_i$ ($i = 1, 2, \ldots, n$) is a collection of HFEs and $u_i$ ($i = 1, 2, \ldots, n$) is the collection of the weights which satisfies the condition (29), then we have the followings:*
*(1) Boundedness: If $h^- = \min\{\gamma_i | \gamma_i \in h_i\}$ and $h^+ = \max\{\gamma_i | \gamma_i \in h_i\}$, then*

$$h^- \leq \mathrm{HFHPOWA}_\zeta(h_1, h_2, \ldots, h_n) \leq h^+.$$

*(2) Monotonicity: Let $h_i'$ ($i = 1, 2, \ldots, n$) be a collection of HFEs, if for any $h_{\sigma(i)}$ and $h_{\sigma(i)}'$ ($i = 1, 2, \ldots, n$), $\gamma_{\sigma(i)} \leq \gamma_{\sigma(i)}'$ , then*

$$\text{HFHPOWA}_\zeta(h_1, h_2, \ldots, h_n) \leq \text{HFHPWA}_\zeta(h_1', h_2', \ldots, h_n').$$

**Definition 10.** *Let $h_i$ ($i = 1, 2, \ldots, n$) be a collection of HFEs. A hesitant fuzzy Hamacher power-ordered weighted geometric (HFHPOWG) operator is a function $H^n \rightarrow H$ such that*

$$\text{HFHPOWG}_\zeta(h_1, h_2, \ldots, h_n) = \otimes_{Hi=1}^n \left( h_{\sigma(i)}^{\wedge_H u_i} \right), \tag{33}$$

*where parameter $\zeta > 0$, $h_{\sigma(i)}$ is the ith largest HFE of $h_j$ ($j = 1, 2, \ldots, n$), and $u_i$ ($i = 1, 2, \ldots, n$) is a collection of weights satisfying the condition (29).*

**Theorem 15.** *If $h_i$ ($i = 1, 2, \ldots, n$) is a collection of HFEs, then the aggregated value by HFHPOWG operator is also an HFE, and*

$$\text{HFHPOWG}_\zeta(h_1, h_2, \ldots, h_n)$$

$$= \cup_{\gamma_{\sigma(1)} \in h_{\sigma(1)}, \gamma_{\sigma(2)} \in h_{\sigma(2)}, \ldots, \gamma_{\sigma(n)} \in h_{\sigma(n)}} \left\{ \frac{\zeta \prod_{i=1}^n \gamma_{\sigma(i)}^{u_i}}{\prod_{i=1}^n \left( 1 + (\zeta - 1)(1 - \gamma_{\sigma(i)}) \right)^{u_i} + (\zeta - 1) \prod_{i=1}^n \gamma_{\sigma(i)}^{u_i}} \right\}, \tag{34}$$

*where $h_{\sigma(i)}$ is the ith largest HFE of $h_j$ ($j = 1, 2, \ldots, n$) and $u_i$ ($i = 1, 2, \ldots, n$) is the collection of the weights satisfying the condition (29).*

**Remark 6.** *(1) If $\text{Sup}(h_i, h_j) = k$, for all $i \neq j$, and $g(x) = x$, then*

$$\text{HFHPOWG}_\zeta(h_1, h_2, \ldots, h_n) = \otimes_{Hi=1}^n \left( h_i^{\wedge_H \frac{1}{n}} \right)$$

*which indicates that the HFHPOWG operator reduces to the hesitant fuzzy Hamacher geometric (HFHG) operator [17].*

*(2) If $\zeta = 1$, then the HFHPOWG operator (34) reduces to the hesitant fuzzy power-ordered weighted geometric (HFPOWG) operator [15]:*

$$\text{HFHPOWG}_1(h_1, h_2, \ldots, h_n) = \otimes_{i=1}^n \left( h_{\sigma(i)} \right)^{u_i}$$

$$= \cup_{\gamma_{\sigma(1)} \in h_{\sigma(1)}, \gamma_{\sigma(2)} \in h_{\sigma(2)}, \ldots, \gamma_{\sigma(n)} \in h_{\sigma(n)}} \left\{ \prod_{i=1}^n \gamma_{\sigma(i)}^{u_i} \right\}, \tag{35}$$

*where $u_i$ ($i = 1, 2, \ldots, n$) is a collection of weights satisfying the condition (29). If $\zeta = 2$, then the HFHPOWG operator (34) reduces to the hesitant fuzzy Einstein power-ordered weighted geometric (HFEPOWG) operator [18]:*

$$\text{HFHPOWG}_2(h_1, h_2, \ldots, h_n)$$

$$= \cup_{\gamma_{\sigma(1)} \in h_{\sigma(1)}, \gamma_{\sigma(2)} \in h_{\sigma(2)}, \ldots, \gamma_{\sigma(n)} \in h_{\sigma(n)}} \left\{ \frac{2 \prod_{i=1}^n \gamma_{\sigma(i)}^{u_i}}{\prod_{i=1}^n \left( 2 - \gamma_{\sigma(i)} \right)^{u_i} + \prod_{i=1}^n \gamma_{\sigma(i)}^{u_i}} \right\}, \tag{36}$$

*where $u_i$ ($i = 1, 2, \ldots, n$) is a collection of weights satisfying the condition (29).*

Similar to Theorems 6–8, we have the properties of HFHPOWG operator as follows.

**Theorem 16.** *If $h_i$ $(i = 1, 2, \ldots, n)$ is a collection of HFEs and $u_i$ $(i = 1, 2, \ldots, n)$ is the collection of the weights which satisfies the condition* (29)*, then*

$$\text{HFHPOWG}_\zeta(h_1, h_2, \ldots, h_n) \geq \text{HFPOWG}(h_1, h_2, \ldots, h_n).$$

**Theorem 17.** *If $h_i$ $(i = 1, 2, \ldots, n)$ is a collection of HFEs and $u_i$ $(i = 1, 2, \ldots, n)$ is the collection of the weights which satisfies the condition* (29)*, then we have the followings:*
　　*(1) Boundedness: If $h^- = \min\{\gamma_i | \gamma_i \in h_i\}$ and $h^+ = \max\{\gamma_i | \gamma_i \in h_i\}$, then*

$$h^- \leq \text{HFHPOWG}_\zeta(h_1, h_2, \ldots, h_n) \leq h^+.$$

　　*(2) Monotonicity: Let $h'_i$ $(i = 1, 2, \ldots, n)$ be a collection of HFEs, if for any $h_{\sigma(i)}$ and $h'_{\sigma(i)}$ $(i = 1, 2, \ldots, n)$, $\gamma_{\sigma(i)} \leq \gamma'_{\sigma(i)}$ , then*

$$\text{HFHPOWG}_\zeta(h_1, h_2, \ldots, h_n) \leq \text{HFHPOWG}_\zeta(h'_1, h'_2, \ldots, h'_n).$$

**Theorem 18.** *If $h_i$ $(i = 1, 2, \ldots, n)$ is a collection of HFEs and $u_i$ $(i = 1, 2, \ldots, n)$ is the collection of the weights satisfying the condition* (29)*, then we have*
　　*(1) $\text{HFHPOWA}_\zeta(h_1^c, h_2^c, \ldots, h_n^c) = (\text{HFHPOWG}_\zeta(h_1, h_2, \ldots, h_n))^c$;*
　　*(2) $\text{HFHPOWG}_\zeta(h_1^c, h_2^c, \ldots, h_n^c) = (\text{HFHPOWA}_\zeta(h_1, h_2, \ldots, h_n))^c$.*

In what follows, we define the generalized hesitant fuzzy Hamacher power-ordered weighted average (GHFHPOWA) operator and generalized hesitant fuzzy Hamacher power-ordered weighted geometric (GHFHPOWG) operator.

**Definition 11.** *Let $h_i$ $(i = 1, 2, \ldots, n)$ be a collection of HFEs. For $\lambda > 0$, a generalized hesitant fuzzy Hamacher power-ordered weighted average (GHFHPOWA) operator is a function $H^n \to H$ such that*

$$\text{GHFHPOWA}_\zeta(h_1, h_2, \ldots, h_n) = \left( \oplus_{H i=1}^{n} \left( u_i \cdot_H h_{\sigma(i)}^{\wedge_H \lambda} \right) \right)^{\wedge_H \frac{1}{\lambda}}, \tag{37}$$

*where $\zeta > 0$, $h_{\sigma(i)}$ is the ith largest HFE of $h_j$ $(j = 1, 2, \ldots, n)$, and $u_i$ $(i = 1, 2, \ldots, n)$ is a collection of weights satisfying the condition* (29)*.*

**Theorem 19.** *Let $h_i$ $(i = 1, 2, \ldots, n)$ be a collection of HFEs, then the aggregated value by GHFHPOWA operator is also an HFE, and*

$$\text{GHFHPOWA}_\zeta(h_1, h_2, \ldots, h_n)$$

$$= \cup_{\gamma_{\sigma(1)} \in h_{\sigma(1)}, \gamma_{\sigma(2)} \in h_{\sigma(2)}, \ldots, \gamma_{\sigma(n)} \in h_{\sigma(n)}} \left\{ \frac{\zeta \left( \prod_{i=1}^{n} a_i^{u_i} - \prod_{i=1}^{n} b_i^{u_i} \right)^{\frac{1}{\lambda}}}{\left[ \begin{array}{c} \left( \prod_{i=1}^{n} a_i^{u_i} + (\zeta^2 - 1) \prod_{i=1}^{n} b_i^{u_i} \right)^{\frac{1}{\lambda}} \\ + (\zeta - 1) \left( \prod_{i=1}^{n} a_i^{u_i} - \prod_{i=1}^{n} b_i^{u_i} \right)^{\frac{1}{\lambda}} \end{array} \right]} \right\}, \tag{38}$$

*where $a_i = (1 + (\zeta - 1)(1 - \gamma_{\sigma(i)})^\lambda + (\zeta^2 - 1)\gamma_{\sigma(i)}^\lambda$, $b_i = (1 + (\zeta - 1)(1 - \gamma_{\sigma(i)}))^\lambda - \gamma_{\sigma(i)}^\lambda$ and $u_i$ $(i = 1, 2, \ldots, n)$ is a collection of weights satisfying the condition* (29)*.*

**Remark 7.** *(1) If $\lambda = 1$, then $a_i = \zeta(1 + (\zeta - 1)\gamma_{\sigma(i)})$ and $b_i = \zeta(1 - \gamma_{\sigma(i)})$, and thus the GHFHPOWA operator reduces to the HFHPOWA operator. In fact, by Equation ([38]), we have*

$$\text{GHFHPOWA}_\zeta(h_1, h_2, \ldots, h_n)$$

$$= \cup_{\gamma_{\sigma(1)} \in h_{\sigma(1)}, \gamma_{\sigma(2)} \in h_{\sigma(2)}, \ldots, \gamma_{\sigma(n)} \in h_{\sigma(n)}} \left\{ \frac{\zeta \left( \prod_{i=1}^n a_i^{u_i} - \prod_{i=1}^n b_i^{u_i} \right)^{\frac{1}{\lambda}}}{\left[ \begin{array}{c} \left( \prod_{i=1}^n a_i^{u_i} + (\zeta^2 - 1) \prod_{i=1}^n b_i^{u_i} \right)^{\frac{1}{\lambda}} \\ + (\zeta - 1) \left( \prod_{i=1}^n a_i^{u_i} - \prod_{i=1}^n b_i^{u_i} \right)^{\frac{1}{\lambda}} \end{array} \right]} \right\}$$

$$= \cup_{\gamma_{\sigma(1)} \in h_{\sigma(1)}, \gamma_{\sigma(2)} \in h_{\sigma(2)}, \ldots, \gamma_{\sigma(n)} \in h_{\sigma(n)}} \left\{ \frac{\prod_{i=1}^n a_i^{u_i} - \prod_{i=1}^n b_i^{u_i}}{\prod_{i=1}^n a_i^{u_i} + (\zeta - 1) \prod_{i=1}^n b_i^{u_i}} \right\}$$

$$= \cup_{\gamma_{\sigma(1)} \in h_{\sigma(1)}, \gamma_{\sigma(2)} \in h_{\sigma(2)}, \ldots, \gamma_{\sigma(n)} \in h_{\sigma(n)}} \left\{ \frac{\prod_{i=1}^n \left( 1 + (\zeta - 1)\gamma_{\sigma(i)} \right)^{u_i} - \prod_{i=1}^n \left( 1 - \gamma_{\sigma(i)} \right)^{u_i}}{\prod_{i=1}^n \left( 1 + (\zeta - 1)\gamma_{\sigma(i)} \right)^{u_i} + (\zeta - 1) \prod_{i=1}^n \left( 1 - \gamma_{\sigma(i)} \right)^{u_i}} \right\}$$

$$= \text{HFHPOWA}_\zeta(h_1, h_2, \ldots, h_n).$$

*(2) If $\zeta = 1$, then the GHFHPOWA operator reduces to the generalized hesitant fuzzy power-ordered weighted average (GHFPOWA) operator [15]:*

$$\text{GHFHPOWA}_1(h_1, h_2, \ldots, h_n) = \cup_{\gamma_{\sigma(1)} \in h_{\sigma(1)}, \gamma_{\sigma(2)} \in h_{\sigma(2)}, \ldots, \gamma_{\sigma(n)} \in h_{\sigma(n)}} \left\{ \left( 1 - \prod_{i=1}^n \left( 1 - \gamma_{\sigma(i)}^\lambda \right)^{u_i} \right)^{\frac{1}{\lambda}} \right\}, \quad (39)$$

*where $u_i$ $(i = 1, 2, \ldots, n)$ is a collection of weights satisfying the condition ([29]), and if $\zeta = 2$, then the GHFHPOWA operator reduces to the generalized hesitant fuzzy Einstein power-ordered weighted average (GHFEPOWA) operator [18]:*

$$\text{GHFHPOWA}_2(h_1, h_2, \ldots, h_n)$$

$$= \cup_{\gamma_{\sigma(1)} \in h_{\sigma(1)}, \gamma_{\sigma(2)} \in h_{\sigma(2)}, \ldots, \gamma_{\sigma(n)} \in h_{\sigma(n)}} \left\{ \frac{2 \left( \prod_{i=1}^n a_i^{u_i} - \prod_{i=1}^n b_i^{u_i} \right)^{\frac{1}{\lambda}}}{\left( \prod_{i=1}^n a_i^{u_i} + 3 \prod_{i=1}^n b_i^{u_i} \right)^{\frac{1}{\lambda}} + \left( \prod_{i=1}^n a_i^{u_i} - \prod_{i=1}^n b_i^{u_i} \right)^{\frac{1}{\lambda}}} \right\}, \quad (40)$$

*where $a_i = (2 - \gamma_{\sigma(i)})^\lambda + 3\gamma_{\sigma(i)}^\lambda$, $b_i = (2 - \gamma_{\sigma(i)})^\lambda - \gamma_{\sigma(i)}^\lambda$ and $u_i$ $(i = 1, 2, \ldots, n)$ is a collection of weights satisfying the condition ([29]).*

**Definition 12.** *Let $h_i$ $(i = 1, 2, \ldots, n)$ be a collection of HFEs. For $\lambda > 0$, a generalized hesitant fuzzy Hamacher power-ordered weighted geometric (GHFHPOWG) operator is a function $H^n \to H$ such that*

$$\text{GHFHPOWG}_\zeta(h_1, h_2, \ldots, h_n) = \frac{1}{\lambda} \cdot_H \left( \otimes_{Hi=1}^n \left( \left( \lambda \cdot_H h_{\sigma(i)} \right)^{\wedge_H u_i} \right) \right), \quad (41)$$

*where $\zeta > 0$, $h_{\sigma(i)}$ is the ith largest HFE of $h_j$ $(j = 1, 2, \ldots, n)$, and $u_i$ $(i = 1, 2, \ldots, n)$ is a collection of weights satisfying the condition ([29]).*

**Theorem 20.** *Let $h_i$ $(i = 1, 2, \ldots, n)$ be a collection of HFEs, then the aggregated value by GHFHPOWG operator is also an HFE, and*

$$\text{GHFHPOWG}_\zeta(h_1, h_2, \ldots, h_n)$$

$$= \cup_{\gamma_{\sigma(1)} \in h_{\sigma(1)}, \gamma_{\sigma(2)} \in h_{\sigma(2)}, \ldots, \gamma_{\sigma(n)} \in h_{\sigma(n)}} \left\{ \frac{\left[ \begin{array}{c} \left(\prod_{i=1}^n c_i^{u_i} + (\zeta^2 - 1) \prod_{i=1}^n d_i^{u_i}\right)^{\frac{1}{\lambda}} \\ - \left(\prod_{i=1}^n c_i^{u_i} - \prod_{i=1}^n d_i^{u_i}\right)^{\frac{1}{\lambda}} \end{array} \right]}{\left[ \begin{array}{c} \left(\prod_{i=1}^n c_i^{u_i} + (\zeta^2 - 1) \prod_{i=1}^n d_i^{u_i}\right)^{\frac{1}{\lambda}} \\ + (\zeta - 1) \left(\prod_{i=1}^n c_i^{u_i} - \prod_{i=1}^n d_i^{u_i}\right)^{\frac{1}{\lambda}} \end{array} \right]} \right\}, \quad (42)$$

*where $c_i = (1 + (\zeta - 1)\gamma_{\sigma(i)})^\lambda + (\zeta^2 - 1)(1 - \gamma_{\sigma(i)})^\lambda$, $d_i = (1 + (\zeta - 1)\gamma_{\sigma(i)})^\lambda - (1 - \gamma_{\sigma(i)})^\lambda$ and $u_i$ $(i = 1, 2, \ldots, n)$ is a collection of weights satisfying the condition (29).*

**Remark 8.** *(1) If $\lambda = 1$, then $c_i = \zeta(1 + (\zeta - 1)(1 - \gamma_{\sigma(i)}))$ and $d_i = \zeta\gamma_{\sigma(i)}$, and the GHFHPOWG operator reduces to the HFHPOWG operator.*

*(2) If $\zeta = 1$, then the GHFHPOWG operator reduces to the generalized hesitant fuzzy power-ordered weighted geometric (GHFPOWG) operator [15]:*

$$\text{GHFHPOWG}_1(h_1, h_2, \ldots, h_n) = \cup_{\gamma_{\sigma(1)} \in h_{\sigma(1)}, \gamma_{\sigma(2)} \in h_{\sigma(2)}, \ldots, \gamma_{\sigma(n)} \in h_{\sigma(n)}} \left\{ \left( \prod_{i=1}^n \left( \gamma_{\sigma(i)}^\lambda \right)^{u_i} \right)^{\frac{1}{\lambda}} \right\}, \quad (43)$$

*where $u_i$ $(i = 1, 2, \ldots, n)$ is a collection of weights satisfying the condition (29), and if $\zeta = 2$, then the GHFHPOWG operator reduces to the generalized hesitant fuzzy Einstein power-ordered weighted geometric (GHFEPOWG) operator [18]:*

$$\text{GHFHPOWG}_2(h_1, h_2, \ldots, h_n)$$

$$= \cup_{\gamma_{\sigma(1)} \in h_{\sigma(1)}, \gamma_{\sigma(2)} \in h_{\sigma(2)}, \ldots, \gamma_{\sigma(n)} \in h_{\sigma(n)}} \left\{ \frac{\left(\prod_{i=1}^n c_i^{u_i} + 3 \prod_{i=1}^n d_i^{u_i}\right)^{\frac{1}{\lambda}} - \left(\prod_{i=1}^n c_i^{u_i} - \prod_{i=1}^n d_i^{u_i}\right)^{\frac{1}{\lambda}}}{\left(\prod_{i=1}^n c_i^{u_i} + 3 \prod_{i=1}^n d_i^{u_i}\right)^{\frac{1}{\lambda}} + \left(\prod_{i=1}^n c_i^{u_i} - \prod_{i=1}^n d_i^{u_i}\right)^{\frac{1}{\lambda}}} \right\}, \quad (44)$$

*where $c_i = (1 + \gamma_{\sigma(i)})^\lambda + 3(1 - \gamma_{\sigma(i)})^\lambda$, $d_i = (1 + \gamma_{\sigma(i)})^\lambda - (1 - \gamma_{\sigma(i)})^\lambda$, and $u_i$ $(i = 1, 2, \ldots, n)$ is a collection of weights satisfying the condition (29).*

## 4. Method for Multiple-Attribute Decision-Making Based on Hesitant Fuzzy Hamacher Power-Aggregation Operators

In this section, we use hesitant fuzzy Hamacher power-aggregation operators to develop an approach to MADM with hesitant fuzzy information.

Let $X = \{x_1, x_2, \ldots, x_n\}$ be a set of $n$ alternatives, and $G = \{g_1, g_2, \ldots, g_m\}$ be a set of $m$ attributes, whose weight vector is $w = (w_1, w_2, \ldots, w_m)^T$, satisfying $w_i > 0$ $(i = 1, 2, \ldots, m)$ and $\sum_{i=1}^m w_i = 1$, where $w_i$ denotes the importance degree of the attribute $g_i$. Suppose the group of decision-makers provides the evaluating value that the alternative $x_j$ $(i = 1, 2, \ldots, n)$ satisfies the attribute $g_i$ $(j = 1, 2, \ldots, m)$ represented by the HFEs $h_{ij}$ $(i = 1, 2, \ldots, m; j = 1, 2, \ldots, n)$. All these HFEs are contained in the hesitant fuzzy decision matrix $D = (h_{ij})_{m \times n}$.

The following steps can be used to solve the MADM problem under the hesitant fuzzy environment, and obtain an optimal alternative:

*Step 1:* Obtain the normalized hesitant fuzzy decision matrix. In general, the attribute set $G$ can be divided two subsets: $G_1$ and $G_2$, where $G_1$ and $G_2$ are the set of benefit attributes and cost attributes, respectively. If all the attributes are of the same type, then the evaluation values do not need normalization, whereas if there are benefit attributes and cost attributes in MADM, in such cases,

we may transform the evaluation values of cost type into the evaluation values of the benefit type by the following normalization formula:

$$r_{ij} = \begin{cases} h_{ij}, & j \in G_1 \\ h_{ij}^c, & j \in G_2, \end{cases} \tag{45}$$

where $h_{ij}^c = \cup_{\gamma_{ij} \in \bar{h}_{ij}} \{1 - \gamma_{ij}\}$ is the complement of $h_{ij}$. Then we obtain the normalized hesitant fuzzy decision matrix $H = (r_{ij})_{m \times n}$.

   *Step 2:* Calculate the supports

$$\text{Sup}(r_{ij}, r_{kj}) = 1 - d(r_{ij}, r_{kj}), \ j = 1, 2, \ldots, n, \ i, k = 1, 2, \ldots, m, \tag{46}$$

which satisfy conditions (1)–(3) in Definition 5. Here we assume that $d(r_{ij}, r_{kj})$ is the hesitant normalized Hamming distance between $r_{ij}$ and $r_{kj}$ given in Equation (4).

   *Step 3:* Calculate the weights of evaluating values. Use the weights $w_i$ ($i = 1, 2, \ldots, m$) of attributes $g_i$ ($i = 1, 2, \ldots, m$) to calculate the weighted support $T(r_{ij})$ of the HFE $r_{ij}$ by the other HFEs $r_{kj}$ ($k = 1, 2, \ldots, m$, and $k \neq i$):

$$T(r_{ij}) = \sum_{k=1, k \neq i}^{m} w_k \text{Sup}(r_{ij}, r_{kj}) \tag{47}$$

and then use the weights $w_i$ ($i = 1, 2, \ldots, m$) of attributes $g_i$ ($i = 1, 2, \ldots, m$) to calculate the weights $\rho_{ij}$ ($i = 1, 2, \ldots, m$) that are associated with HFEs $r_{ij}$ ($i = 1, 2, \ldots, m$):

$$\rho_{ij} = \frac{w_i(1 + T(r_{ij}))}{\sum_{i=1}^{m} w_i(1 + T(r_{ij}))}, \ i = 1, 2, \ldots, m, \tag{48}$$

where $\rho_{ij} \geq 0$, $i = 1, 2, \ldots, m$, and $\sum_{i=1}^{m} \rho_{ij} = 1$.

   *Step 4:* Compute overall assessments of alternatives. Use the HFHPWA operator (Equation (6)):

$$
\begin{aligned}
r_j &= \text{HFHPWA}_\zeta(h_1, h_2, \ldots, h_n) \\
&= \cup_{\gamma_{1j} \in r_{1j}, \gamma_{2j} \in r_{2j}, \ldots, \gamma_{mj} \in r_{mj}} \left\{ \frac{\prod_{i=1}^{m}(1 + (\zeta - 1)\gamma_{ij})^{\rho_{ij}} - \prod_{i=1}^{m}(1 - \gamma_{ij})^{\rho_{ij}}}{\prod_{i=1}^{m}(1 + (\zeta - 1)\gamma_{ij})^{\rho_{ij}} + (\zeta - 1)\prod_{i=1}^{m}(1 - \gamma_{ij})^{\rho_{ij}}} \right\},
\end{aligned} \tag{49}
$$

or the HFHPWG operator (Equation (11)):

$$
\begin{aligned}
r_j &= \text{HFHPWG}_\zeta(h_1, h_2, \ldots, h_n) \\
&= \cup_{\gamma_{1j} \in r_{1j}, \gamma_{2j} \in r_{2j}, \ldots, \gamma_{mj} \in r_{mj}} \left\{ \frac{\zeta \prod_{i=1}^{m}(\gamma_{ij})^{\rho_{ij}}}{\prod_{i=1}^{m}(1 + (\zeta - 1)(1 - \gamma_{ij}))^{\rho_{ij}} + (\zeta - 1)\prod_{i=1}^{m}(\gamma_{ij})^{\rho_{ij}}} \right\}. \tag{50}
\end{aligned}
$$

to aggregate all the evaluating values $\bar{r}_{ij}$ ($1 = 1, 2, \ldots, m$) of the $j$th column and get the overall rating value $\bar{r}_j$ corresponding to the alternative $x_j$ ($j = 1, 2, \ldots, n$).

   *Step 5:* Rank the order of all alternatives. Use the method in Definition 3 to rank the overall rating values $r_j$ ($j = 1, 2, \ldots, n$), rank all the alternatives $x_j$ ($j = 1, 2, \ldots, n$) in accordance with $r_j$ ($j = 1, 2, \ldots, n$) in descending order, and finally select the most desirable alternative(s) with the largest overall evaluation value.

   *Step 6:* End.

**Remark 9.** *As previously discussed, a family of hesitant fuzzy Hamacher power-aggregation operators, including the HFHPWA, HFHPWG, GHFHPWA, GHFHPWG, HFHPOWA, HFHPOWG, GHFHPOWA, and GHFHPOWG operators, is proposed for aggregating hesitant fuzzy information. This family is composed*

*of two kinds: the HFHPWA operator and HFHPWG operator, and other aggregation operators are developed based on them. Therefore, in Step 3, the HFHPWA and HFHPWG operators are chosen to aggregate hesitant fuzzy information.*

**Example 4.** *There is a five-member board of directors of a company. They plan to invest their money in a suitable project with a lot of potential over the next five years [17]. Assume that the board of directors will evaluate the four possible projects $X = \{x_1, x_2, x_3, x_4\}$. To evaluate and rank these projects, four attributes $G = \{g_1, g_2, g_3, g_4\}$ are suggested by the Balances Score Card Methodology, i.e., (1) $g_1$ is the financial perspective; (2) $g_2$ is the customer satisfaction; (3) $g_3$ is the internal business process perspective; and (4) $g_4$ is the learning and growth perspective. Please note that these attributes are all benefit attributes and the corresponding weight vector is $w = (0.2, 0.3, 0.15, 0.35)^T$. The five members of the board of directors provide the evaluating values of the projects $x_j$ (j = 1, 2, 3, 4) with respect to attributes $g_i$ (i = 1, 2, 3, 4) and construct their hesitant fuzzy decision matrix $D = (h_{ij})_{4 \times 4}$ (see Table 1), where $h_{ij} \in H$ is a HFE that denotes all of the possible values for alternative $x_j$ under the attribute $g_i$.*

**Table 1.** Hesitant fuzzy decision matrix $D$.

|  | $x_1$ | $x_2$ | $x_3$ | $x_4$ |
|---|---|---|---|---|
| $g_1$ | $\{0.2, 0.4, 0.7\}$ | $\{0.2, 0.4, 0.7, 0.9\}$ | $\{0.3, 0.5, 0.6, 0.7\}$ | $\{0.3, 0.5, 0.6\}$ |
| $g_2$ | $\{0.2, 0.6, 0.8\}$ | $\{0.1, 0.2, 0.4, 0.5\}$ | $\{0.2, 0.4, 0.5, 0.6\}$ | $\{0.2, 0.4\}$ |
| $g_3$ | $\{0.2, 0.3, 0.6, 0.7, 0.9\}$ | $\{0.3, 0.4, 0.6, 0.9\}$ | $\{0.3, 0.5, 0.7, 0.8\}$ | $\{0.5, 0.6, 0.7\}$ |
| $g_4$ | $\{0.3, 0.4, 0.5, 0.7, 0.8\}$ | $\{0.5, 0.6, 0.8, 0.9\}$ | $\{0.2, 0.5, 0.6, 0.7\}$ | $\{0.8, 0.9\}$ |

*Then we use the above proposed approach to choose the optimal project.*

*Step 1: Since these attributes are all benefit attributes, it is not necessary to normalize the decision matrix D.*

*Step 2: Use Equation(46) to calculate the supports $\mathrm{Sup}(h_{ij}, h_{kj})$ (j = 1, 2, 3, 4, i, k = 1, 2, 3, 4, i ≠ k). For simplicity, we denote $(\mathrm{Sup}(h_{ij}, h_{kj}))_{1 \times 4}$ by $\mathrm{Sup}_{ik}$, which refers to the supports between the ith and kth columns of D:*

$$\mathrm{Sup}_{12} = \mathrm{Sup}_{21} = (0.900, 0.750, 0.900, 0.800), \ \mathrm{Sup}_{13} = \mathrm{Sup}_{31} = (0.800, 0.950, 0.950, 0.867),$$

$$\mathrm{Sup}_{14} = \mathrm{Sup}_{41} = (0.800, 0.850, 0.975, 0.633), \ \mathrm{Sup}_{23} = \mathrm{Sup}_{32} = (0.860, 0.750, 0.850, 0.667),$$

$$\mathrm{Sup}_{24} = \mathrm{Sup}_{42} = (0.860, 0.600, 0.925, 0.495), \ \mathrm{Sup}_{34} = \mathrm{Sup}_{43} = (0.920, 0.850, 0.925, 0.767).$$

*Step 3: Use Equation (47) to calculate the weighted support $T(h_{ij})$ of HFE $h_{ij}$ by the other HFEs $h_{kj}$ (k = 1, 2, 3, 4, k ≠ i), which are contained in the matrix $T = (T(h_{ij}))_{4 \times 4}$:*

$$T = \begin{pmatrix} 0.6700 & 0.6650 & 0.7538 & 0.5916 \\ 0.6100 & 0.4725 & 0.6313 & 0.4333 \\ 0.7400 & 0.7125 & 0.7688 & 0.6420 \\ 0.5560 & 0.4775 & 0.6113 & 0.3902 \end{pmatrix}$$

*and use Equation (48) to calculate the weights $\rho_{ij}$ of HFEs $h_{ij}$ (i = 1, 2, 3, 4), which are contained in the matrix $V = (\rho_{ij})_{4 \times 4}$:*

$$V = \begin{pmatrix} 0.2058 & 0.2150 & 0.2101 & 0.2149 \\ 0.2977 & 0.2852 & 0.2932 & 0.2903 \\ 0.1609 & 0.1659 & 0.1589 & 0.1663 \\ 0.3356 & 0.3339 & 0.3378 & 0.3285 \end{pmatrix}.$$

*Step 4: Let $\zeta = 0.5$ and use the HFHPWA operator (Equation (49)) to aggregate all of the evaluating values $h_{ij}$ (i = 1, 2, 3, 4) in the jth column of D and then, derive the overall rating value $h_j$ (j = 1, 2, 3, 4) of the*

*alternative $x_j$ ($j = 1, 2, 3, 4$). The overall rating values $h_j$ are not listed here because of limited space. Using Definition 3, we calculate the score functions $s(h_j)$ of $h_j$ ($j = 1, 2, 3, 4$) as follows:*

$$s(h_1) = 0.5741, s(h_2) = 0.6195, s(h_3) = 0.5256, s(h_4) = 0.6646.$$

*Then we rank the $h_j$ ($j = 1, 2, 3, 4$) in descending order of $s(h_j)$:*

$$h_4 > h_2 > h_1 > h_3.$$

*Step 5: Rank all the alternatives $x_j$ ($j = 1, 2, 3, 4$) as follows:*

$$x_4 \succ x_2 \succ x_1 \succ x_3.$$

*Thus, the best alternative is $x_4$.*

*Furthermore, let $\zeta = 0.1, 0.3, 0.5, 3, 5, 7$, respectively, which represents different preferences for decision-makers on decision information. We can obtain the corresponding score values and rankings of the alternatives (listed in Table 2).*

**Table 2.** Score values obtained with the HFHPWA operator and rankings of alternatives.

| Aggregation Operator | Score Values | Rankings |
|---|---|---|
| HFHPWA$_{0.1}$ | $s(h_1) = 0.5957, s(h_2) = 0.6501, s(h_3) = 0.5318, s(h_4) = 0.7000$ | $x_4 \succ x_2 \succ x_1 \succ x_3$ |
| HFHPWA$_{0.3}$ | $s(h_1) = 0.5826, s(h_2) = 0.6314, s(h_3) = 0.5298, s(h_4) = 0.6780$ | $x_4 \succ x_2 \succ x_1 \succ x_3$ |
| HFHPWA$_{0.5}$ | $s(h_1) = 0.5741, s(h_2) = 0.6195, s(h_3) = 0.5256, s(h_4) = 0.6646$ | $x_4 \succ x_2 \succ x_1 \succ x_3$ |
| HFHPWA$_3$ | $s(h_1) = 0.5419, s(h_2) = 0.5752, s(h_3) = 0.5063, s(h_4) = 0.6186$ | $x_4 \succ x_2 \succ x_1 \succ x_3$ |
| HFHPWA$_5$ | $s(h_1) = 0.5350, s(h_2) = 0.5653, s(h_3) = 0.5017, s(h_4) = 0.6095$ | $x_4 \succ x_2 \succ x_1 \succ x_3$ |
| HFHPWA$_7$ | $s(h_1) = 0.5313, s(h_2) = 0.5598, s(h_3) = 0.4993, s(h_4) = 0.6047$ | $x_4 \succ x_2 \succ x_1 \succ x_3$ |

*To explain how the different parameter value $\zeta$ plays a role in the aggregation operator, we use the different values $\zeta$, given by decision-makers. As shown in Table 2, the score values obtained by the HPHPWA operator become smaller as the parameter value $\zeta$ increases. Thus, decision-makers can choose the parameter value $\zeta$ according to their preferences.*

**Table 3.** Score values obtained with the HFHPWG operator and rankings of alternatives.

| Aggregation Operator | Score Values | Rankings |
|---|---|---|
| HFHPWG$_{0.1}$ | $s(h_1) = 0.4397, s(h_2) = 0.4118, s(h_3) = 0.4187, s(h_4) = 0.4673$ | $x_4 \succ x_1 \succ x_3 \succ x_2$ |
| HFHPWG$_{0.3}$ | $s(h_1) = 0.4520, s(h_2) = 0.4321, s(h_3) = 0.4373, s(h_4) = 0.4824$ | $x_4 \succ x_1 \succ x_3 \succ x_2$ |
| HFHPWG$_{0.5}$ | $s(h_1) = 0.4603, s(h_2) = 0.4444, s(h_3) = 0.4483, s(h_4) = 0.4930$ | $x_4 \succ x_1 \succ x_3 \succ x_2$ |
| HFHPWG$_3$ | $s(h_1) = 0.4935, s(h_2) = 0.4936, s(h_3) = 0.4904, s(h_4) = 0.5410$ | $x_4 \succ x_2 \succ x_1 \succ x_3$ |
| HFHPWG$_5$ | $s(h_1) = 0.5009, s(h_2) = 0.5027, s(h_3) = 0.5005, s(h_4) = 0.5536$ | $x_4 \succ x_2 \succ x_1 \succ x_3$ |
| HFHPWG$_7$ | $s(h_1) = 0.5049, s(h_2) = 0.5126, s(h_3) = 0.5061, s(h_4) = 0.5608$ | $x_4 \succ x_2 \succ x_3 \succ x_1$ |

*If the HFHPWA operator is replaced by HFHPWG operator in the above Step 4, Table 3 lists the score values of overall rating values and rankings of alternatives. The score values obtained by the HFHPWG operator become larger as parameter $\zeta$ increases. Comparing Table 2 with Table 3, we can observe that the score value obtained by the HFHPWA operator greater than the score value obtained by the HFHPWG operator for the same parameter value $\zeta$ and the same aggregation value. For HFHPWA operators, the parameter value does not affect the final rankings of the alternatives, but for the HFHPWG operators it is shown that the choice of parameter value has a greater impact on the score values and thus the rankings of the alternatives. In the above analysis, we see that while the best alternative obtained by the HFHPWA operator are the same as that obtained by the HFHPWG operator, the rankings of the alternatives differs between the HFHPWA and HFHPWG operators.*

To compare our approach with some other approaches, we apply Xia and Xu's approach [10], Zhu et al.'s approach [13], and Tan et al.'s approach [17] to above example. The results of the rankings of the alternatives are shown in Table 4 below.

**Table 4.** Rankings of alternatives by other approaches.

| Approach | Use Tool | Rankings |
|----------|----------|----------|
| Xia and Xu [10] | HFWA operator | $x_4 \succ x_2 \succ x_1 \succ x_3$ |
| | HFWG operator | $x_4 \succ x_1 \succ x_3 \succ x_2$ |
| Zhu et al. [13] | Hesitant fuzzy TOPSIS | $x_4 \succ x_2 \succ x_1 \succ x_3$ |
| Tan et al. [17] | HFHWA operator | $x_4 \succ x_2 \succ x_1 \succ x_3$ |
| | HFHWG operator | $x_4 \succ x_3 \succ x_1 \succ x_2 \ (\zeta \in (0, 0.36])$ |
| | | $x_4 \succ x_1 \succ x_3 \succ x_2 \ (\zeta \in (0.36, 1.39])$ |
| | | $x_4 \succ x_1 \succ x_2 \succ x_3 \ (\zeta \in (1.39, 3.50])$ |
| | | $x_4 \succ x_2 \succ x_1 \succ x_3 \ (\zeta \in (3.50, 10.0])$ |

From this analysis, we can see that the best alternative is the same for the both HFHPWA and HFHPWG operators, or both HFWA and HFWG operators, or both HFHWA and HFHWG operators, but the ranking of alternatives is different between the HFHPWA and HFHPWG operators, or HFWA and HFWG operators, or HFHWA and HFHWG operators. It reflects that the final results may be different by different types of hesitant fuzzy aggregation operators. Also, the ranking of alternatives obtained by the hesitant fuzzy TOPSIS method is the same those by the HFHPWA, HFWA, and HFHWA operators. This result shows the validity of the proposed approach in this paper.

Compared with the existing hesitant fuzzy MADM approaches, our proposed approach has two advantages: First, decision-makers often have an optimistic or pessimistic attitude in the face of decision information. In this case, optimistic attitude often leads to a preference for risk-seeking, and pessimistic one results in a preference for avoiding risk. The parameter $\zeta$ takes into account the decision-maker's subjective attitude to decision-making problem and are therefore useful in obtaining a better decision result. Second, different parameter values clearly indicate changes in the ranking of alternatives. Compared to a fixed evaluated result obtained by existing aggregation operators such as the HFWA and HFWG operators, our evaluated result can better reflect the variety.

## 5. Conclusions

Hesitant fuzzy information aggregation is one of key issues in the hesitant fuzzy MADM, an important field of research in decision science in an uncertain environment as well as HFS theory. Based on Hamacher operations of HFEs, in this paper, we have developed a family of hesitant fuzzy Hamacher power-aggregation operators, including the HFHWPA, HFHPWG, GHFHPWA, GHFHPWG, HFHPOWA, HFHPOWG, GHFHPOWA, and GHFHPOWG operators. Some basic properties of the proposed aggregation operators, such as boundedness and monotonicity, and the relationships between them have been investigated and discussed. We compared the proposed aggregation operators with the hesitant fuzzy aggregation operators developed by Yu et al. [18] and Zhang [15] and represented their corresponding relations. These proposed hesitant fuzzy Hamacher power-aggregation operators are integrated treatment of operators proposed by Yu et al. [18] and Zhang [15], and provide a complement to the existing work on HFSs. An approach of the hesitant fuzzy MADM based on the HFHPWA and HFHPWG operators has been developed and an example of money investment selection has been provided to describe the hesitant fuzzy MADM process. Some advantages of our proposed approach are shown by comparison with those previously proposed by Xia and Xu [10], Zhu et al. [13] and Tan et al. [17].

In future work, we will present a series of hesitant fuzzy power-aggregation operators using Frank *t*-norm and *t*-conorm and apply them to develop approaches for multiple-attribute group

decision-making. Furthermore, we will discuss the extension of power-aggregation operators to probabilistic hesitant fuzzy environment.

**Author Contributions:** J.H.P. drafted the initial manuscript and conceived the MADM framework. M.J.S. provided the relevant literature review and illustrated example. M.J.S. and K.H.K. revised the manuscript and analyzed the data.

**Funding:** This work was supported by a Research Grant of Pukyong National University (2019).

**Conflicts of Interest:** The authors declare no conflict of interest.

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
