# Peer review of "Some Hesitant Fuzzy Hamacher Power-Aggregation Operators for Multiple-Attribute Decision-Making"

_mathematics, doi:10.3390/math7070594_

Round 1

Reviewer 1 Report

The article is well written and give an interesting conclusions. It is based on a correct mathematic apparatus, the line of reasoning is clear and intelligible, and most pieces of information necessary to follow the line of reasoning is contained in the text. The authors presented an comprehensive description of the proposed approach, prepared an analysis of the sensitivity of solving the decision problem and compared their own approach with others use in literature. Nonetheless, some minor elements of the article ought to be improved.

Primarily, the authors should carefully check the paper for grammar and typos errors (eg. line 5 "propoerties", line 35 "So these operators have received a lot of attention from researchers in recent years, and Xu and Yager [24]...", line 74 "and and").

Secondly, the method is based on the hesitant fuzzy set. In my opinion, one paragraph in the introduction should be devoted to discuss more classical approaches and applications to fuzzy multi-criteria decision making and their applications. Some recent articles dealing with this issue can be mentioned, e.g.:

- Wang, C.-N.; Yang, C.-Y.; Cheng, H.-C. Fuzzy Multi-Criteria Decision-Making Model for Supplier Evaluation and Selection in a Wind Power Plant Project. Mathematics 2019, 7, 417. https://doi.org/10.3390/math7050417

- Ziemba, P.; Becker, J. Analysis of the Digital Divide Using Fuzzy Forecasting. Symmetry 2019, 11, 166. https://doi.org/10.3390/sym11020166

- Mari, S.I.; Memon, M.S.; Ramzan, M.B.; Qureshi, S.M.; Iqbal, M.W. Interactive Fuzzy Multi Criteria Decision Making Approach for Supplier Selection and Order Allocation in a Resilient Supply Chain. Mathematics 2019, 7, 137. https://doi.org/10.3390/math7020137

- Wang, T.-C.; Tsai, S.-Y. Solar Panel Supplier Selection for the Photovoltaic System Design by Using Fuzzy Multi-Criteria Decision Making (MCDM) Approaches. Energies 2018, 11, 1989. https://doi.org/10.3390/en11081989

 - Jankowski, J.; Wątróbski, J. Online Comparison System with Certain and Uncertain Criteria Based on Multi-criteria Decision Analysis Method. Lecture Notes in Computer Science 2017, 10449, 579-589.  https://doi.org/10.1007/978-3-319-67077-5_56

- Polomčić, D.; Gligorić, Z.; Bajić, D.; Gligorić, M.; Negovanović, M. Multi-Criteria Fuzzy-Stochastic Diffusion Model of Groundwater Control System Selection. Symmetry 2019, 11, 705. https://doi.org/10.3390/sym11050705

- Jaini, N.I.; Utyuzhnikov, S.V. A Fuzzy Trade-Off Ranking Method for Multi-Criteria Decision-Making. Axioms 2018, 7, 1. https://doi.org/10.3390/axioms7010001

Author Response

Repsponses to Reviewer 1 comments:

1. We thank the reviewer for pointing out this issue. In this paper, we further studied the Hamacher operation rules on HFEs, defined a variety of power aggregation operators based on these operations, discussed their properties, and compared them to the existing aggregation operators on HFEs. Then, the MADM problem solution based on these operators was developed and compared to the existing MADM methods.

According to reviewer's advice, As reviewer's comments, we added some papers on MCDM/MADM applied to various regions in References, and based on that, we revised the introduction as follows.

- The goal of multiple attribute decision making (MADM), based on preferences provided by the decision makers, is to select the most desirable alternative(s) from a given set of feasible alternatives. MADM methods classified as conventional and fuzzy. The conventional MADM methods are seen inadequate to handle uncertainty in linguistic terms [23]. Hence, it is proposed to apply MADM methods with the FS and its extensions to cope with vagueness in a decision making process. Furthermore, these fuzzy methods enable to obtain more concrete results. Besides, the FS and its extensions helps to decision makers in order to express their opinions by means of linguistic terms. Therefore, more sensitive results can be obtained by applying fuzzy MADM methods to various science and engineering fields such as supplier selection and forecasting [24-27].

2. As reviewer's comments, we carefully checked and amended the paper for grammar and typos errors.

Reviewer 2 Report

This work proposes some new hesitant fuzzy Hamacher power aggregation operators for hesitant fuzzy information based on Hamacher t-norm and t-conorm. Some basic propoerties of these operators is given and the relationships between them are shown. Further the interrelations between the proposed aggregation operators and the existing hesitant fuzzy power aggregation operators are discussed. Based on the proposed aggregation operators, the authors develop a new approach for multiple attribute decision making problems. Finally, a practical example is provided to illustrate the effectiveness of the proposed approach, and the advantages of the proposed approach by comparing with other existing approach are analyzed. The ideas in the paper are interesting and the theoretic results obtained have good potential in applications. This paper can be accepted for publication after some necessary changes: (1) The presentation and language quality should be further improved; (2) Some remark words on the computation complexity of the obtained results should be given. In particular, the efficiency on the implementation issue should be highlighted; (3) The unique features of the approaches proposed and the main advantages of the results over others have to be clearly commented, especially in the simulation section; (4) Some future works should be proposed in the conclusion section; (5) The notations used in the paper should be refined; (6) Please give a short discussion if it is possible to extend the current results for the potential modelling and control of nonlinear dynamic systems and for this issue, , the following paper can be included to improve the literature review: observer-based fuzzy adaptive event-triggered control for pure-feedback nonlinear systems with prescribed performance, IEEE TFS, doi: 10.1109/TFUZZ.2019.2895560; (7) It would be much better if some guideline remark words on practical applications should be given; (8) The paper also has some typos and language issue which need to be checked and corrected in the revision. Overall, the paper is interesting and can be accepted for publication after fully consideration of aforementioned issues.

Author Response

Responses to Reviewer 2:

1. As reviewers comments, we carefully checked and amended the paper for grammar and
typos errors and present the issues in the revision version of the paper.

2. As reviewer’s comments, we described the main benefits of the results for the proposed
approach and the other approach in Example 4 as follows:
- From this analysis, we can see that the best alternative is the same for the both
HFHPWA and HFHPWG operators, or both HFWA and HFWG operators, or both
HFHWA and HFHWG operators, but the ranking of alternatives is different between the
HFHPWA and HFHPWG operators, or HFWA and HFWG operators, or HFHWA and
HFHWG operators. It reflects that the final results may be different by different types of
hesitant fuzzy aggregation operators. Also, the ranking of alternatives obtained by the
hesitant fuzzy TOPSIS method is the same those by the HFHPWA, HFWA and HFHWA
operators. This result shows the validity of the proposed approach in this paper.
Compared with the existing hesitant fuzzy MADM approaches, our proposed approach
has two advantages: First, decision makers often have an optimistic or pessimistic at�titude in the face of decision information. In this case, optimistic attitude often leads
to a preference for risk-seeking, and pessimistic one results in a preference for avoiding
risk. The parameter ζ takes into account the decision maker’s subjective attitude to
decision making problem and are therefore useful in obtaining a better decision result.
Second, different parameter values clearly indicate changes in the ranking of alternatives.
Compared to a fixed evaluated result obtained by existing hesitant fuzzy aggregation op�erators such as the HFWA and HFWG operators, the our evaluated result can better
reflect the variety.

3. As reviewer’s advice, we proposed some future works in the conclusion section as follows:
- In future work, we will present a series of hesitant fuzzy power aggregation operators
using Frank t-norm and t-conorm and apply them to develop approaches for multiple
attribute group decision making. Furthermore, we will discuss the extension of power
aggregation operators to probabilistic hesitant fuzzy environment.

4. The reviewer proposed to outline the modeling and control of nonlinear dynamic systems
and where current results on this issue can be extended. I agree with the reviewer’
commernt and this will apply our results to the potential modeling of nonlinear dynamic
systems as a future project.

Round 2

Reviewer 2 Report

no further comments.